# Tracking receptor motions at the plasma membrane reveals distinct effects of ligands on CCR5 dynamics depending on its dimerization status

**Fanny Momboisse[1], Giacomo Nardi[2], Philippe Colin[3], Melanie Hery[1], Nelia Cordeiro[1], Simon Blachier[4], Olivier Schwartz[1], Fernando Arenzana-Seisdedos[5], Nathalie Sauvonnet[6], Jean-Christophe Olivo-Marin[2], Bernard Lagane[3], Thibault Lagache[2]\*, Anne Brelot[1]\***

[1]Institut Pasteur, Université Paris Cité, CNRS UMR3569, Virus and Immunity Unit, Paris, France; [2]Institut Pasteur, Université Paris Cité, CNRS UMR3691, BioImage Analysis Unit, Paris, France; [3]Infinity, Université de Toulouse, CNRS, INSERM, Toulouse, France; [4]Institut Pasteur, Université Paris Cité, Dynamics of Host-Pathogen Interactions Unit, Paris, France; [5]Institut Pasteur, Université Paris Cité, INSERM U1108, Viral Pathogenesis Unit, Paris, France; [6]Institut Pasteur, Université Paris Cité, Group Intracellular Trafficking and Tissue Homeostasis, Paris, France

**\*For correspondence:**
thibault.lagache@pasteur.fr (TL);
anne.brelot@pasteur.fr (AB)

**Abstract** G-protein-coupled receptors (GPCR) are present at the cell surface in different conformational and oligomeric states. However, how these states impact GPCRs biological function and therapeutic targeting remains incompletely known. Here, we investigated this issue in living cells for the CC chemokine receptor 5 (CCR5), a major receptor in inflammation and the principal entry co-receptor for Human Immunodeficiency Viruses type 1 (HIV-1). We used TIRF microscopy and a statistical method to track and classify the motion of different receptor subpopulations. We showed a diversity of ligand-free forms of CCR5 at the cell surface constituted of various oligomeric states and exhibiting transient Brownian and restricted motions. These forms were stabilized differently by distinct ligands. In particular, agonist stimulation restricted the mobility of CCR5 and led to its clustering, a feature depending on β-arrestin, while inverse agonist stimulation exhibited the opposite effect. These results suggest a link between receptor activation and immobilization. Applied to HIV-1 envelope glycoproteins gp120, our quantitative analysis revealed agonist-like properties of gp120s. Distinct gp120s influenced CCR5 dynamics differently, suggesting that they stabilize different CCR5 conformations. Then, using a dimerization-compromised mutant, we showed that dimerization (i) impacts CCR5 precoupling to G proteins, (ii) is a pre-requisite for the immobilization and clustering of receptors upon activation, and (iii) regulates receptor endocytosis, thereby impacting the fate of activated receptors. This study demonstrates that tracking the dynamic behavior of a GPCR is an efficient way to link GPCR conformations to their functions, therefore improving the development of drugs targeting specific receptor conformations.

## Editor's evaluation

This manuscript seeks to push the frontiers of live-cell single-molecule imaging by tracking the diffusive movements of CCR5 receptors and CCR5 receptor complexes within the plasma membrane of living cells and how these motional behaviors change with physiological stimuli. The results will

be important for researchers working at the interface of cell biology and biophysics on membrane-bound receptors.

## Introduction

G-protein-coupled receptors (GPCRs), also known as 7TM (seven transmembrane helical) receptors, represent the largest group of cell surface receptors in humans that transduce chemical signals from the extracellular matrix into the cell. They constitute one of the primary drug target classes (*Pierce et al., 2002*).

GPCRs exist in different subpopulations at the cell surface, in part due to differential post-translational modifications (*Patwardhan et al., 2021*; *Scurci et al., 2021*) and arrangements of receptor loops and transmembrane domains (*Deupi and Kobilka, 2010*). Receptor activation and G protein coupling indeed involves a series of conformational changes from an inactive to an active state (*Ahn et al., 2021*). Coupling to different G proteins or to other protein transducers (e.g. arrestins), as well as receptor oligomerization expand the diversity of conformational states (*Seyedabadi et al., 2019*; *Sleno and Hébert, 2018*). Molecular dynamics along with biophysical and structural studies brought to light this variety of GPCR arrangements and showed how binding of different ligands can stabilize or select different receptor conformations, which can in turn activate different signaling pathways (*Ahn et al., 2021*). This concept of 'functional selectivity' (or 'biased agonism') opens the possibility to develop therapies specifically targeting a selected receptor conformation, thereby increasing the effectiveness of drugs and reducing their adverse effects (*Seyedabadi et al., 2019*).

The nature and proportion of the different forms of GPCRs vary depending on their environment. This is likely to regulate the functional properties of the receptors (*Colin et al., 2018*; *Patwardhan et al., 2021*). Few studies, however, confirmed this diversity of receptors in living cells and investigated its regulation in time and space (*Calebiro et al., 2012*; *Gormal et al., 2020*; *Kasai et al., 2018*; *Martínez-Muñoz et al., 2018*; *Sungkaworn et al., 2017*; *Veya et al., 2015*). In this study, we tracked the chemokine receptor CCR5 at the particle level to access its dynamic behavior at the plasma membrane and identify the organization and the functional properties of the various receptor forms.

CCR5 is a class A GPCR expressed on the surface of hematopoietic and non-hematopoietic cells. It is a key player in the trafficking of lymphocytes and monocytes/macrophages and has been implicated in the pathophysiology of multiple diseases, including viral infections and complex disorders with an inflammatory component (*Brelot and Chakrabarti, 2018*; *Flanagan, 2014*; *Vangelista and Vento, 2017*). In addition, the CCL5/CCR5 axis represents a major marker of tumor development (*Aldinucci et al., 2020*). CCR5 binds several chemokines, including CCL3, CCL4, and CCL5. Binding of chemokines results in conformational change of the receptor, which then activates intracellular signaling pathways and leads to cell migration (*Flanagan, 2014*). CCR5 also binds the envelope glycoprotein of HIV-1, then acting as the major HIV-1 entry co-receptor (*Alkhatib et al., 1996*; *Brelot and Chakrabarti, 2018*). One CCR5 allosteric ligand, maraviroc (MVC), is part of the anti-HIV-1 therapeutic arsenal (*Dorr et al., 2005*), although emergence of MVC-resistant variants has been identified in some patients (*Tilton et al., 2010*).

We and others showed the existence of various CCR5 populations present at the cell surface (*Abrol et al., 2014*; *Berro et al., 2011*; *Colin et al., 2013*; *Colin et al., 2018*; *Fox et al., 2015*; *Jacquemard et al., 2021*; *Jin et al., 2014*; *Jin et al., 2018*; *Scurci et al., 2021*). Computational analysis predicts that CCR5 can adopt an ensemble of low-energy conformations, each of which being differentially favored by distinct ligands and receptor mutations (*Abrol et al., 2014*). CCR5 conformations display distinct antigenic properties, which vary depending on cell types (*Colin et al., 2018*; *Fox et al., 2015*). The multiple conformations interact differently with distinct ligands (agonist, antagonist, HIV-1 envelope glycoprotein) and differ in their biological properties, HIV co-receptor functions, and abilities to serve as therapeutic targets (*Abrol et al., 2014*; *Colin et al., 2013*; *Colin et al., 2018*; *Jacquemard et al., 2021*; *Jin et al., 2014*; *Jin et al., 2018*; *Scurci et al., 2021*). In particular, coupling to G proteins distinguishes CCR5 populations that are differently engaged by chemokines and HIV-1 envelope. This explains why HIV-1 escapes inhibition by chemokines (*Colin et al., 2013*). In this context, the improved capacity of chemokine analogs to inhibit HIV infection, as compared to native chemokines, is related to their ability to target a large amount of CCR5 conformations (*Jin et al., 2014*).

Like other receptors of this class, CCR5 forms homo- and heterodimers with other receptors, which contribute to the diversity of conformational states (*Jin et al., 2018*; *Sohy et al., 2009*). We identified three homodimeric organizations of CCR5 involving residues of transmembrane domain 5 (TM5) (*Jin et al., 2018*). Two dimeric states corresponded to unliganded receptors, whereas binding of the inverse agonist MVC stabilized a third state (*Jin et al., 2018*). CCR5 dimerization occurs in the endoplasmic reticulum, thereby regulating the receptor targeting to the cell surface (*Jin et al., 2018*). CCR5 dimerization also modulates ligand binding and HIV-1 entry into cells (*Colin et al., 2018*). MVC stabilizes CCR5 homodimerization, illustrating that CCR5 dimerization can be modulated by ligands (*Jin et al., 2018*), a feature shared with other chemokine receptors (*Işbilir et al., 2020*). Allosteric interaction within CCR2/CCR5 heterodimers is reported as well as cross-inhibition by specific antagonists (*Sohy et al., 2009*). This suggests that dimerization impacts therapeutic targeting.

To characterize the diversity of CCR5 subpopulations at the cell surface and to investigate the impact of CCR5 dynamics on its function, we tracked CCR5 fluorescent particles by total internal reflection fluorescence (TIRF) microscopy (*Calebiro et al., 2012*) and quantitatively classify their motion over time using a statistical method. We described CCR5 mobility patterns both at the basal state and after ligand binding (using two agonists, the inverse agonist MVC, and HIV-1 envelope glycoproteins) and under conditions that modulate CCR5 /G protein coupling, β-arrestin binding, and dimerization. This study provides novel insights into the organization of a GPCR at the cell surface and the mechanisms regulating its signaling and fate after activation.

## Results

### Statistical classification of receptor trajectories at the cell membrane

We studied CCR5 dynamics in two different models: eGFP-CCR5 and FLAG-SNAP-tagged-CCR5 (FLAG-ST-CCR5) expressing cells, in which we tracked either eGFP or receptor-bound fluorescent anti-FLAG antibodies. We used HEK 293 cell lines stably expressing a low density of eGFP-CCR5 or FLAG-ST-CCR5 at the cell surface (<0.5 particles/µm²), which is critical for single particle tracking on the surface of living cells (*Calebiro et al., 2012*). We chose HEK 293 cells because they do not express CCR5. Fusion of proteins to the N-terminus of CCR5 does not alter cell surface expression of the receptor or its intracellular trafficking (*Boncompain et al., 2019*; *Jin et al., 2018*).

To study the dynamics of CCR5 as a single particle at the plasma membrane of living cells, we used TIRF microscopy, which restricts the observation to the first 200 nm from the coverslip. The acquisitions were carried out at 37 °C. From the movies obtained, we tracked the motion of the particles over time using the *Spot tracking* plugin of the ICY software (*Chenouard et al., 2013*; *de Chaumont et al., 2012*; *Figure 1A–C*, *Videos 1–4*, see Materials and methods).

The method generally used to evaluate the dynamics of a particle is based on Mean Square Displacement (MSD) analysis (*Qian et al., 1991*). However, MSD is a global analysis of particle trajectory that does not handle possible changes in particle motion. In particular, it indicates whether the observed motion is standard Brownian motion and computes the related diffusion coefficient of the trajectory, but it cannot characterize more complex stochastic motions as the frequency of motion changes. In addition, the MSD analysis does not provide a statistical significance of classified motion. More robust analysis using the Bayesian probabilistic framework have been proposed to classify single particle trajectories (*Karslake et al., 2021*; *Monnier et al., 2015*; *Türkcan and Masson, 2013*).

However, Bayesian inference is often associated with a high computational load and is not very robust for short trajectories. Therefore, to robustly characterize the complex stochastic motions of single receptors at the cell membrane, we chose to implement a statistical hypothesis testing method introduced in *Briane et al., 2018*. To mitigate the risk of tracking errors over long trajectories, and to detect potential motion changes between tracklets within each single particle trajectory, we partitioned single spot trajectories into small tracklets (with N=5 consecutive detections each; *Figure 1D* and *Figure 1—figure supplement 1A*). We first evaluated immobile objects and then used a robust statistical method to classify tracklet motion (see Materials and methods and *Figure 1E–F*). Briefly, for each tracklet $X$, we computed the statistics $S(X, N)$ introduced in *Briane et al., 2018* that evaluate the ratio between the maximal distance reached by the tracklet particle from the initial point and the motion standard deviation. We then used the statistics $S(X, N)$ to classify each tracklet into one of the three following motion categories: *confined*, *Brownian*, or *directed* stochastic motion.

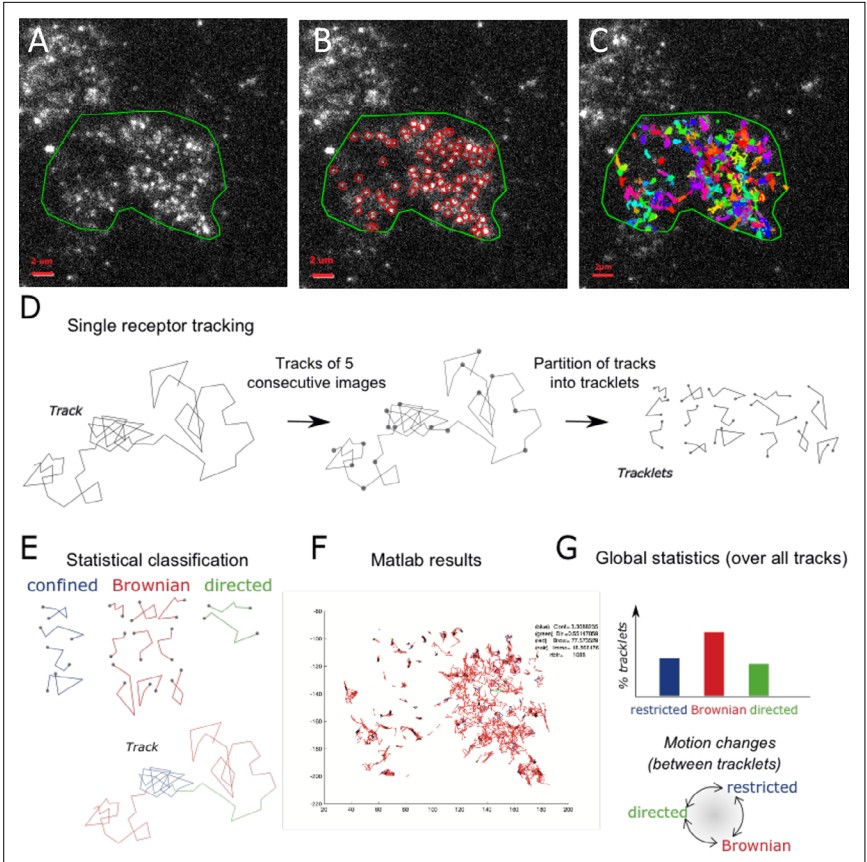

**Figure 1.** Single particle detection of eGFP-CCR5 using TIRF microscopy and analysis with the statistical method. (**A**) Distribution of eGFP-CCR5 stably expressed in HEK 293 cells. Imaging was acquired at 30 Hz. The region of interest defined by the green line is used for A-C and F. Analysis of movies was performed using the ICY software and (**B**) the *Spot detection* and (**C**) the *Spot tracking* plugins. Scale bar 2 μm. (**D**) Single receptor tracks were partitioned into tracklets of five images each. (**E**) Analysis of tracks with the statistical method: tracklets were classified into confined, Brownian, and directed motion. (**F**) Results obtained from Matlab. (**G**) Pooled tracklets classification provided a global estimate of receptor dynamics and the number of motion changes along the track (transition rates). (Restricted motions: immobile and confined motions).

The online version of this article includes the following source data, source code, and figure supplement(s) for figure 1:

**Source code 1.** Matlab code used for simulations.

**Figure supplement 1.** Validation of the statistical classification method using simulated trajectories and synthetic time-lapse sequences.

**Figure supplement 1—source data 1.** Source data for *Figure 1—figure supplement 1*.

For this, we computed *S(X,N)* for each tracklet and compared it to the quantiles $(q_\alpha, q_{1-\alpha})$, which are statistical reference values of Brownian motion at level α and $(1 - \alpha)$. Quantiles of *S(X,N)* only depend on *N* and $\alpha$ (*Briane et al., 2018*), and can be evaluated independently of the characteristics of experimental trajectories. Finally, tracklets $X$ were classified according to the associated stochastic motion: confined (if $S(X,N) < q(\alpha)$), Brownian (if $q(\alpha) \le S(X,N) < q(1-\alpha)$), and directed motion (if $q(1-\alpha) \le S(X,N)$ (*Figure 1—figure supplement 1*)). Finally, to evaluate the robustness of tracklet classification to image noise and receptors' density, we generated synthetic time-lapse sequences (Materials and methods) and measured the classification accuracy for different signal-to-noise ratio (from SNR = 2 to SNR = 10) and receptors' spots density = 0.039, 0.16 and 0.63 spots/$\mu m^2$, the measured density being < 0.5 $spots/\mu m^2$ in most experiments. Our simulations showed that classification accuracy was maintained for SNR >6 (*Figure 1—figure supplement 1B*), the experimental SNR being ~10, and that classification was robust to spots' density (*Figure 1—figure supplement 1C*).

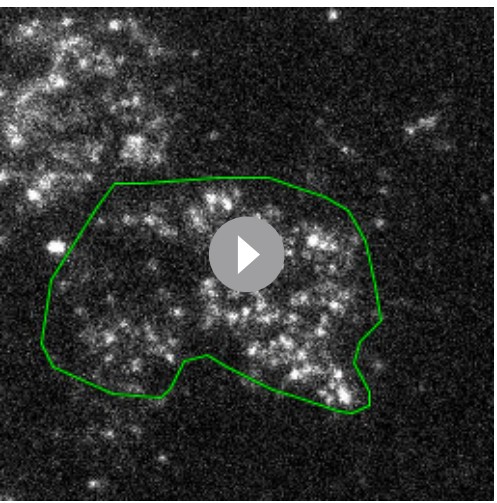

**Video 1.** TIRF movie of a cell stably expressing eGFP-CCR5-WT acquired at 30 Hz. The region of interest was defined by the green line.

https://elifesciences.org/articles/76281/figures#video1

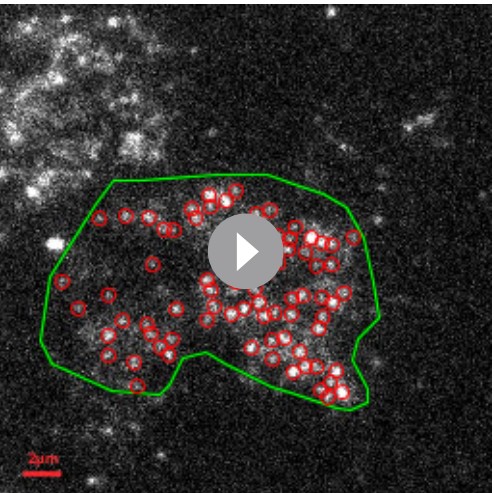

**Video 2.** TIRF movie of the same cell as in *Video 1* analyzed using the Icy software. Red circles correspond to the detection of bright spots using the *Spot detection* plugin.

https://elifesciences.org/articles/76281/figures#video2

After having implemented this statistical classification in the ICY software (processor *Dynamics Classifier* in the plugin *Track Manager*), we characterized the dynamics of CCR5 particles at the cell membrane.

## CCR5 particles have different motions at the plasma membrane

We investigated CCR5 mobility in the basal state using the statistical method described above (*Figure 1*). The result of the classification of all the pooled tracklets provided a global estimate of the receptor dynamics, while the number of motion changes along the same trajectory gave us an estimate of the overall stability of the motion (*Figure 1G*).

In the basal state, the eGFP-CCR5 particles distributed homogeneously over the entire membrane surface (*Figure 1A*, *Videos 1–3*). However, the motions of eGFP-CCR5 particles were heterogeneous (*Figure 2A*). Eighty percent of the pooled CCR5 tracklets were mobile with Brownian motion, while 20% were classified as restricted motion (i.e. immobile and confined; *Figure 2A*). We observed almost no directed trajectories (<0.5 %). Around 50% of particles (52%) exhibited Brownian motion over the entire length of the path (*Figure 2B*). The other half fluctuated between Brownian and restricted motion (*Figure 2B*). This high degree of

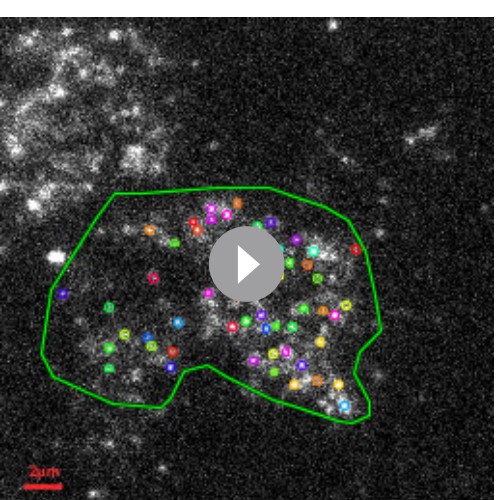

**Video 3.** TIRF movie of the same cell as in *Videos 1 and 2* analyzed using the Icy software and the *Spot tracking* plugin. Colored lines correspond to the tracked spots.

https://elifesciences.org/articles/76281/figures#video3

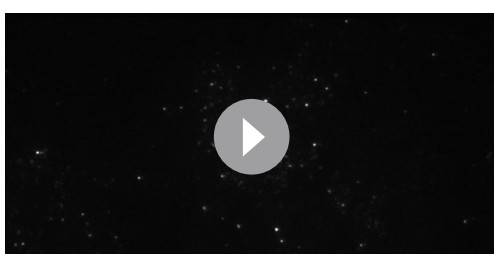

**Video 4.** TIRF movie of a cell stably expressing FLAG-ST-CCR5-WT and stained with M2-Cy3. Movie was acquired at 10 Hz.

https://elifesciences.org/articles/76281/figures#video4

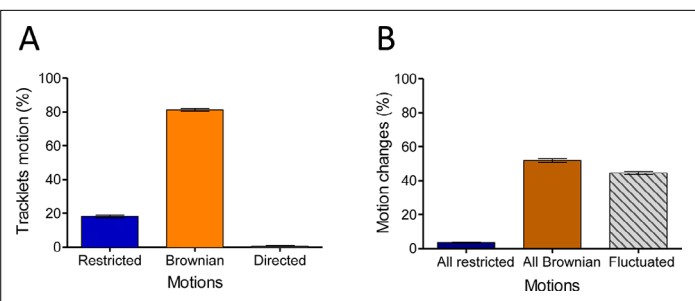

**Figure 2.** In the basal state, eGFP-CCR5 exhibits different motions at the plasma membrane. (**A**) Distribution of tracklets motion: restricted, Brownian, or directed (mean ± SEM, n=28,305 tracks from 19 cells, 3 independent experiments). (**B**) Distribution of tracklets motion changes along tracks (mean ± SEM, n=48,237 tracks from 45 cells, 7 experiments).

The online version of this article includes the following source data and figure supplement(s) for figure 2:

**Source data 1.** Source data for *Figure 2*.

**Figure supplement 1.** In the basal state, FLAG-ST-CCR5 exhibits different motions at the plasma membrane.

**Figure supplement 1—source data 1.** Source data for *Figure 2—figure supplement 1*.

fluctuation between motions within one trajectory suggested the existence of transient conformations of CCR5 at the plasma membrane. Similarly, the motions of FLAG-ST-CCR5 particles were heterogeneous with high degree of fluctuation between motions (*Figure 2—figure supplement 1*). Note that compared to eGFP-CCR5, FLAG-ST-CCR5 exhibited a higher percentage of tracklet in restricted motion (50%), which we attributed to antibody binding (*Harms et al., 2012*).

Together, these analyses revealed heterogeneity of CCR5 motion at the basal state consistent with the diversity of CCR5 forms described previously by other methods (*Abrol et al., 2014*; *Colin et al., 2013*; *Fox et al., 2015*; *Jin et al., 2018*; *Scurci et al., 2021*).

## Multiple ligands impact CCR5 mobility differently

Since ligands modulate the conformation of CCR5 (*Colin et al., 2018*; *Jacquemard et al., 2021*; *Jin et al., 2018*), we investigated the impact of ligand binding on its spatiotemporal dynamic properties. We evaluated the effect of saturating concentration of ligands (two agonists with different efficacies and the inverse agonist MVC, i.e. a ligand with a negative efficacy) on CCR5 trajectories at the plasma membrane over time. We first incubated eGFP-CCR5-expressing cells in the presence of the native CCR5 chemokine CCL4 at a saturating concentration (>100 nM, kd = 0.4 nM; *Colin et al., 2013*) for the indicated time. The mobility of the receptor was then assessed immediately after addition of the ligand in a window of 1–12 min (*Figure 3A*). CCL4 triggered no significant change in CCR5 mobility after 10 min of stimulation (*Figure 3B*). However, a longer time of CCL4 stimulation (>12 min) increased the percentage of restricted CCR5 tracklets, indicating localized immobility of a small fraction of receptors (*Figure 3—figure supplement 1*). We also noted the formation of large and immobile spots after 12 min of stimulation (*Video 5*).

We compared the effect of CCL4 with that of an agonist targeting a greater proportion of receptor conformations and displaying a greater agonist efficacy, PSC-RANTES (*Escola et al., 2010*; *Jin et al., 2014*). We incubated the cells in the presence of a saturating concentration of PSC-RANTES (20 nM, Ki = 1.9 nM; *Colin et al., 2013*) and evaluated the motion of the receptors under the same conditions. PSC-RANTES triggered a progressive increase in the number of tracklets classified as restricted motion over time (*Figure 3A*). Ten minutes after stimulation with PSC-RANTES, about 50% of eGFP-CCR5 tracklets were in a restricted state (46 %) against 17% under basal conditions (*Figure 3B*). Consequently, the fraction of all Brownian trajectories decreased, while the fraction of fluctuated and all restricted trajectories increased (*Figure 3C*). Simultaneously, we observed the formation of large immobile spots (5–10 per cell) in PSC-RANTES-treated cells (*Figure 3D, left*). These large spots had a long lifespan (50–100 frames) (*Video 6*). The quantification of the fluorescence intensity of the spots from the frame 1 of live-imaging movies showed that the large spots had, on average, intensity four times higher than the other spots, indicating a clustering of at least four receptors per large spot

(*Figure 3D, right*). These results revealed a change in CCR5 mobility upon activation toward receptor immobilization and clustering, supporting receptors trapping in nanodomains.

Unlike agonists, the inverse agonist MVC (10 µM, Kd = 1 nM) (*Garcia-Perez et al., 2011*) did not restrict receptor mobility (*Figure 3A, B and C*). On the contrary, the fraction of restricted eGFP-CCR5 tracklets at the surface of MVC-treated cells showed a slight decrease compared to untreated cells (*Figure 3B*). We verified the specificity of PSC-RANTES-induced CCR5 immobility by treating cells with MVC before PSC-RANTES stimulation. MVC treatment impaired PSC-RANTES-induced receptor immobilization (*Figure 3E–F*), indicating that CCR5 immobilization depended on PSC-RANTES binding to CCR5. We observed the same effect of ligand binding (using CCL4, PSC-RANTES, and MVC) on FLAG-ST-CCR5 mobility, supporting that our findings were independent of the model used (*Figure 3—figure supplement 2*).

These results showed that distinct ligands differently stabilize CCR5 in living cells, in accordance with our previous results (*Colin et al., 2013*; *Colin et al., 2018*; *Jin et al., 2014*; *Jin et al., 2018*). Interestingly, the amount of receptors immobilized correlates with the efficacy of ligands (PSC-RANTES >CCL4>MVC), suggesting a link between receptor activation and immobilization.

## Gi coupling and β-arrestin association influence CCR5 motion differently under basal state and stimulated conditions

To further address the above hypothesis, we sought to determine whether the mobility of CCR5 is influenced by its coupling to Gi protein, which stabilizes the receptor in an activated state. We analyzed the pool of restricted CCR5 tracklets in the presence of pertussis toxin (PTX), which uncouples the receptor from Gi proteins (*Figure 4A*).

In the basal state, the fraction of restricted eGFP-CCR5 tracklets from cells pre-treated with PTX decreased compared to untreated cells (*Figure 4A*). Under this condition, PTX also inhibited chemotaxis, a process that depends on CCR5 coupling to Gi proteins (*Figure 4—figure supplement 1*). These results thus suggested that a small subset of CCR5 is in a Gi-protein-bound form in its basal state, which may contribute to the transient restriction of the motion of CCR5 at the cell surface.

After stimulation, receptor immobilization could be due to the recruitment of receptors in hub areas where the receptor meets the activation machinery and in particular the G protein (*Sungkaworn et al., 2017*). To evaluate the role of Gi coupling on receptor immobilization after PSC-RANTES stimulation, we analyzed tracks of TIRF movies of PSC-RANTES-stimulated cells pretreated or not with PTX. In this condition, the fraction of restricted tracklets increased over time after stimulation in the same proportion regardless of PTX treatment (*Figure 4B*). This suggested that Gi coupling was not involved in PSC-RANTES dependent immobilization of CCR5 after several minutes of stimulation. This result is

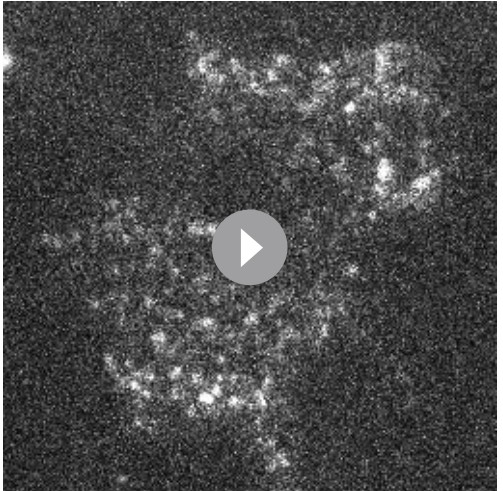

**Video 5.** TIRF movie acquired at 30 Hz of a cell stably expressing eGFP-CCR5-WT and treated by CCL4 (100 nM) for 14 min.
https://elifesciences.org/articles/76281/figures#video5

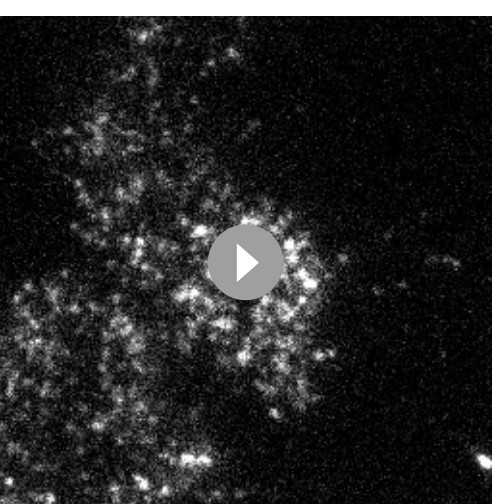

**Video 6.** TIRF movie acquired at 30 Hz of cells stably expressing eGFP-CCR5-WT and treated by PSC-RANTES (20 nM) for 3 min.
https://elifesciences.org/articles/76281/figures#video6

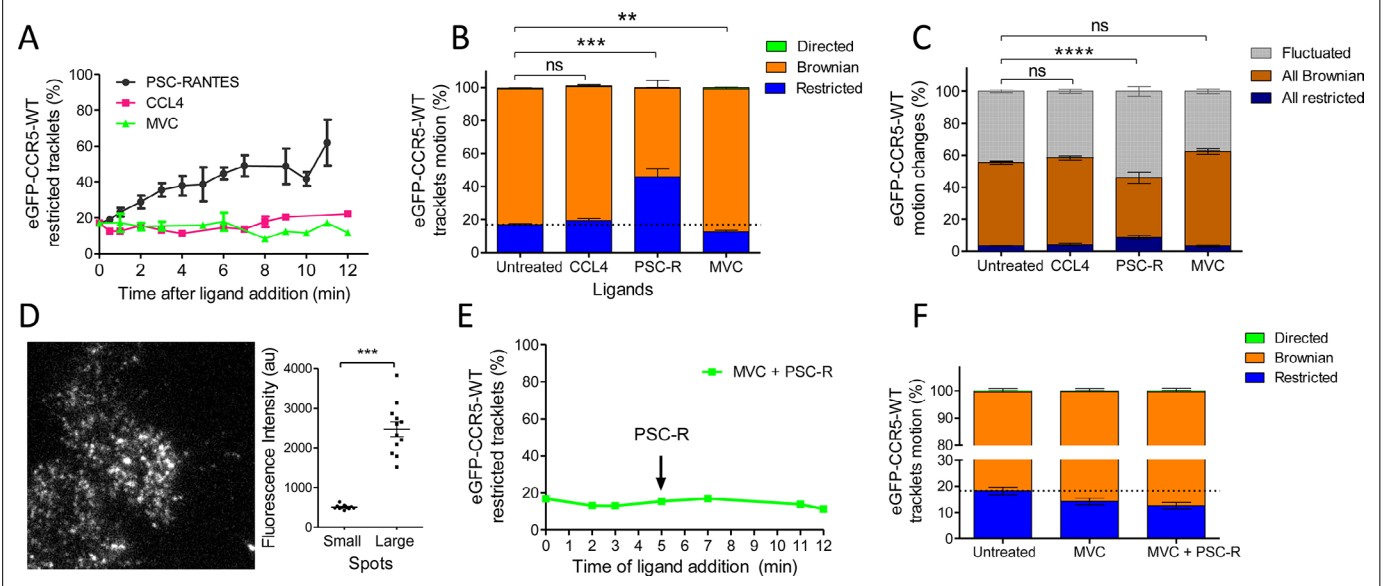

**Figure 3.** Different ligands, agonists and inverse agonist, impact eGFP-CCR5 mobility differently. eGFP-CCR5-WT expressing cells were treated or not with a saturating concentration of agonists (CCL4, 200 nM or PSC-RANTES, 20 nM) or inverse agonist (maraviroc, 10 µM) and single particle tracking analysis was performed. (**A**) Percentage of restricted tracklets after treatment over time (n=tracks from 10, 4, and 3 cells for PSC-RANTES, CCL4, and MVC conditions respectively, at least three independent experiments). (**B**) Distribution of tracklets motion after 10 min of treatment (mean ± SEM, n=40,564, 15,421, 11,213, 9828 tracks for each condition from 35, 12, 12, and 9 cells, respectively, at least three independent experiments). Unpaired t test on restricted motions only: ns, nonsignificant; **p≤0.01; ***p≤0.001. (**C**) Distribution of tracklets motion changes along tracks after 10 min of treatment (mean ± SEM, n=48,237, 8954, 16,668, 9828 tracks from 45, 9, 17, and 9 cells for each condition respectively, at least three experiments). Unpaired t test on all restricted motions only: ns, nonsignificant; ****p≤0.0001. (**D**) (Left) Single particle detection of eGFP-CCR5-WT after 3 min of stimulation with PSC-RANTES (20 nM) from frame 1 of live-imaging movie (one representative image). (Right) Mean of the sum of fluorescence intensity under large immobile spots and small mobile spots after 3–10 min of stimulation (mean ± SEM, n=at least 40 spots from 12 cells, three experiments). (**E**) Percentage of restricted tracklets after successive stimulation with maraviroc (10 µM, 5 min) and PSC-RANTES (20 nM, 5–12 min; one representative experiment). (**F**) Distribution of tracklets motions after successive stimulation with maraviroc (10 µM, during 5 min) and PSC-RANTES (20 nM, during 6 min) (mean ± SEM, n=14,467, 3601, 2075 tracks from 14, 2, and 2 cells respectively, one experiment).

The online version of this article includes the following source data and figure supplement(s) for figure 3:

**Source data 1.** Source data for *Figure 3*.

**Figure supplement 1.** Effect of CCL4 on eGFP-CCR5 mobility.

**Figure supplement 1—source data 1.** Source data for *Figure 3—figure supplements 1 and 2*.

**Figure supplement 2.** Different ligands, agonists and inverse agonist, impact FLAG-ST-CCR5 mobility differently.

actually consistent with our previous study showing high affinity interaction of PSC-RANTES with Gi protein uncoupled CCR5 (*Colin et al., 2013*).

After stimulation by PSC-RANTES, CCR5 follows a clathrin-dependent endocytosis pathway, involving β-arrestins, which bridge the receptor to AP2 and clathrin (*Delhaye et al., 2007*; *Jin et al., 2014*). We previously showed that silencing β-arrestin 1 and β-arrestin 2 endogeneous expressions with siRNA decreased CCR5 internalization after PSC-RANTES stimulation (*Jin et al., 2014*). Silencing β-arrestins in eGFP-CCR5 cells with siRNA did not impact eGFP-CCR5 motion in the basal state (*Figure 4C*) but inhibited PSC-RANTES-induced eGFP-CCR5 immobilization and clustering (*Figure 4D*). These experiments indicated that β-arrestins contributed to CCR5 immobilization after stimulation.

Together, these results pointed to the existence of a fraction of CCR5 in a transient pre-assembled signaling complex in the basal state, which is consistent with previous studies showing CCR5 constitutive activity (*Garcia-Perez et al., 2011*; *Lagane et al., 2005*). They also suggested that the fate of CCR5 several minutes after activation is independent of Gi coupling but dependent on β-arrestin recruitment, in accordance with receptor desensitization and uncoupling after activation (*Flanagan, 2014*).

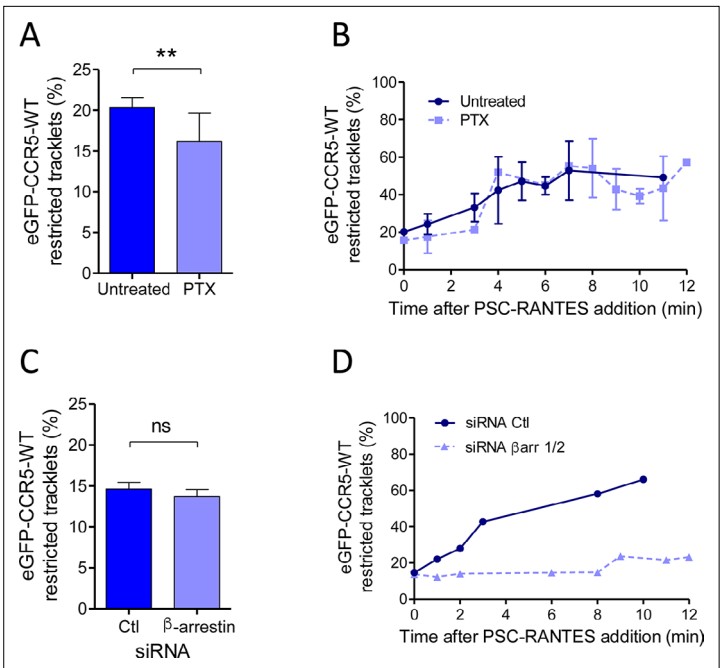

**Figure 4.** Gi coupling and β-arrestins association restrict eGFP-CCR5 mobility at basal state or after PSC-RANTES stimulation. (**A**) Percentage of restricted tracklets in eGFP-CCR5-WT expressing HEK 293 cells pre-treated or not with 100 ng/ml of PTX for 3 hr (mean ± SEM, n=8614 and 11 377 tracks for each condition, 12 and 15 cells respectively, 3 independent experiments). Unpaired t test: p value 0.0083**. (**B**) Percentage of restricted tracklets over time of eGFP-CCR5-WT expressed on PSC-RANTES (20 nM) treated cells after incubation or not with PTX (100 ng/ml) (mean ± SD, n=3 independent experiments). (**C**) Proportion of restricted tracklets in eGFP-CCR5-WT expressing cells transfected with siRNA βarr1/2 (mean ± SD, n=6754 and 8854 tracks for each condition, from 7 and 8 cells, respectively). Unpaired t test: p value 0.46, ns. (**D**) Percentage of restricted tracklets over time of eGFP-CCR5-WT expressed on PSC-RANTES (20 nM) treated cells after siRNA βarr 1/2 transfection (n=1 representative experiment).

The online version of this article includes the following source data and figure supplement(s) for figure 4:

**Source data 1.** Source data for *Figure 4*.

**Figure supplement 1.** Effect of PTX treatment on chemokine-mediated chemotaxis.

**Figure supplement 1—source data 1.** Source data for *Figure 4—figure supplement 1*.

## Immobilization of CCR5 after stimulation depends on its oligomeric state

We previously showed by energy transfer experiments, molecular modeling, and a functional assay that a point mutation of CCR5 in TM5 (L196K) leads to a receptor, which has a reduced dimerization capacity compared to CCR5-WT (*Jin et al., 2018*). Functionally, this mutation alters CCR5 cell surface expression due to its intracellular retention in the endoplasmic reticulum (*Jin et al., 2018*). However, CCR5-L196K folding is not impacted: CCR5-L196K binds chemokines and HIV gp120s with the same affinity as CCR5-WT (*Colin et al., 2018*; *Jin et al., 2018*) and triggered ERK1/2 activation upon stimulation (*Figure 5—figure supplement 1*). To study the role of CCR5 dimerization on its mobility, we generated HEK 293 cells stably expressing eGFP-CCR5-L196K in the same proportion to the clone expressing eGFP-CCR5-WT.

We studied the molecular composition of both eGFP-CCR5-L196K and eGFP-CCR5-WT in these cells by analyzing the fluorescence intensity of eGFP per spot from the frame 1 of live-imaging movies. In a previous study, we calibrated the fluorescence intensity of eGFP while spotted on glass coverslip (*Salavessa et al., 2021*). We showed that most of eGFP spots bleached in a single step, suggesting that eGFP corresponds to 1 molecule, with an average fluorescence intensity of 300–500 au (*Salavessa et al., 2021*). In eGFP-CCR5 expressing cells, the fluorescence intensities were distributed in Gaussians, which we classified with the Akaike information criterion (AIC, see Materials and methods)

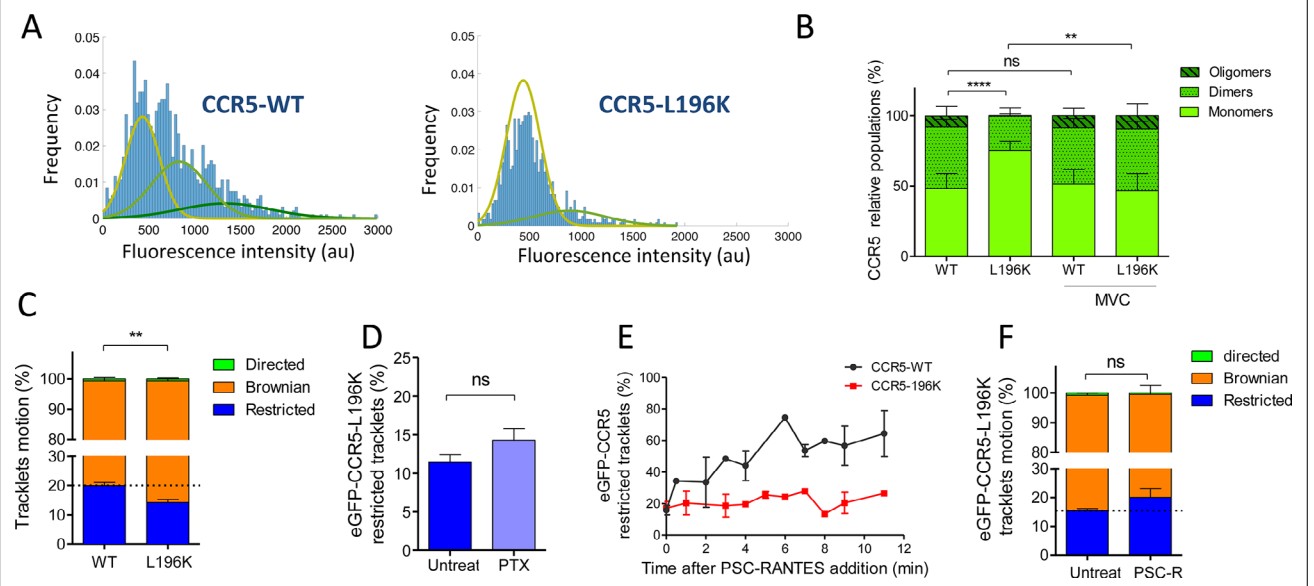

**Figure 5.** Dimerization through TM5 alters eGFP-CCR5 mobility. (**A**) Distribution of the fluorescence intensity of spots detected at the surface of HEK 293 cells expressing eGFP-CCR5-WT or eGFP-CCR5-L196K. One representative experiment out of 6 (n=943 spots from 6 cells and 1207 spots from 8 cells for each condition); (**B**) Quantification of the fluorescent populations depending on the mean of the gaussian at the surface of cells treated or not with MVC (10 µM) (mean ± SD, nWT = 5171 spots from 47 cells, 11 experiments; nL196K=3144 spots from 30 cells, 7 experiments; nWT-MVC=3 055 spots from 25 cells, 4 experiments; nL196K-MVC=1 776 spots from 16 cells, 3 experiments). Unpaired t test on monomers: p value **p≤0.005; ****p≤0.0001; ns p≥0.05; (**C**) Distribution of pooled tracklets motion of eGFP-CCR5-WT and eGFP-CCR5-L196K (mean ± SEM, n=11,321 tracks from 10 cells and 10,460 tracks from 12 cells in each condition; 2 independent experiments). Unpaired t test on the restricted tracklets: p value 0.0015**. (**D**) Percentage of restricted tracklets in eGFP-CCR5-L196K cells pre-treated or not with 100 ng/ml of PTX for 3 hr (mean ± SEM, n=5 cells). Unpaired t test: p value 0.15, ns. (**E**) Percentage of restricted tracklets over time of PSC-RANTES induced eGFP-CCR5-WT or eGFP-CCR5-L196K expressing cells (mean ± SD of 3 independent experiments). (**F**) Distribution of tracklets motion after 10 min of PSC-RANTES stimulation (20 nM) (mean ± SEM, n=11 218 tracks from 10 cells and 5 433 tracks from 4 cells for untreated and PSC-RANTES treated cells respectively, 2 independent experiments). Unpaired t test: p value 0.055, ns.

The online version of this article includes the following source data and figure supplement(s) for figure 5:

**Source data 1.** Source data for *Figure 5*.

**Figure supplement 1.** CCR5-WT and CCR5-L196K promote chemokine-induced ERK activation.

**Figure supplement 1—source data 1.** Source data for *Figure 5—figure supplements 1 and 2*.

**Figure supplement 2.** Dimerization through TM5 alters FLAG-ST-CCR5 mobility.

(*Akaike, 1974*). We observed three types of Gaussians with double or triple mean intensities (300, 600, 900 au), which may correspond to spots comprising 1, 2, or 3 fluorescence entities relative to eGFP on coverslip (*Figure 5A*). This reflected the existence of a heterogeneous distribution of receptors. In this classification, the WT receptor distributed in 50% low, 40% medium, and 10% high fluorescence intensity forms at the plasma membrane, while eGFP-CCR5-L196K was mostly in a low fluorescence intensity form (75% low, 25% medium) (*Figure 5B*). These results revealed that eGFP-CCR5-L196K existed more as monomers or small-size oligomers compared to CCR5-WT at the surface of living cells. This is consistent with the role of Leu-196 in CCR5 oligomerization (*Jin et al., 2018*).

In the presence of MVC, both eGFP-CCR5-WT and eGFP-CCR5-L196K distribution exhibited 50% low, 40% medium, and 10% high fluorescence intensity forms (*Figure 5B*). The change of eGFP-CCR5-L196K fluorescence intensities distribution in the presence of MVC is consistent with our previous results showing that MVC stabilized CCR5 in a novel oligomeric form, which was not disrupted by the introduction of a lysine in TM5 (*Jin et al., 2018*).

To investigate the impact of CCR5 dimerization on its mobility, we compared the motion of eGFP-CCR5-L196K to eGFP-CCR5-WT at the cell surface. As for eGFP-CCR5-WT, eGFP-CCR5-L196K tracklets were predominantly classified as Brownian tracklets motion (85% of the tracklet motions are Brownian). However, we observed a decrease in the proportion of restricted tracklets for eGFP-CCR5-L196K compared to eGFP-CCR5-WT (*Figure 5C*). We observed the same decrease in the proportion

of restricted tracklets for FLAG-ST-CCR5-L196K compared to FLAG-ST-CCR5-WT (*Figure 5—figure supplement 2*). These data suggested that the degree of receptor oligomerization contributed to the stability of CCR5 molecules at the cell surface, as previously proposed (*Calebiro et al., 2012*).

To test whether eGFP-CCR5-L196K coupling to Gi protein accounts in its restriction as shown for eGFP-CCR5-WT, we pre-treated cells with PTX. Contrary to eGFP-CCR5-WT, PTX treatment did not alter the proportion of the eGFP-CCR5-L196K restricted tracklets pool (*Figure 5D*), suggesting that most of eGFP-CCR5-L196K were not precoupled to the Gi protein at the basal state or that G protein precoupling induces differential effects on the dynamics of both receptors. Supporting the first hypothesis, previous biochemical and energy transfer experiments on a distinct GPCR showed that there could be a link between dimerization and Gi coupling at basal state (*Maurice et al., 2010*).

To investigate whether dimerization affected CCR5 mobility after stimulation, we analyzed single-particle movies of eGFP-CCR5-L196K cells after PSC-RANTES treatment (*Figure 5E–F*). Contrary to eGFP-CCR5-WT massive immobilization and clustering upon PSC-RANTES treatment (*Figure 3A–B*), eGFP-CCR5-L196K was only slightly immobilized after 10 min of treatment (*Figure 5E–F*), while large immobile spots were not detected (*Video 7*). This result indicated that CCR5 immobilization and clustering after stimulation depend on CCR5 dimerization.

Because CCR5-WT immobilization involved β-arrestins (*Figure 4D*), an explanation for the lack of PSC-RANTES induced eGFP-CCR5-L196K immobilization is that eGFP-CCR5-L196K fails to recruit β-arrestins and therefore, is not desensitized and/or internalized after stimulation.

To test this hypothesis, we evaluated PSC-RANTES-induced β−arrestin 2 (βarr2) recruitment at the plasma membrane of cells expressing either FLAG-ST-CCR5-L196K or FLAG-ST-CCR5-WT (*Jin et al., 2018*). TIRF acquisitions were performed in fixed cells transiently expressing βarr2-GFP previously stained for FLAG detection. CCR5 activation drove rapid recruitment of βarr2-GFP into spots close to the plasma membrane (*Figure 6A*). The proportion of recruited βarr2-GFP at the plasma membrane was similar for CCR5-WT and CCR5-L196K (*Figure 6B*), suggesting that βarr2 recruitment is independent of the oligomeric status of the receptor. Note that we observed a slight decrease in the number βarr2-GFP spots that colocalize with fluorescent receptor spots in CCR5-L196K expressing

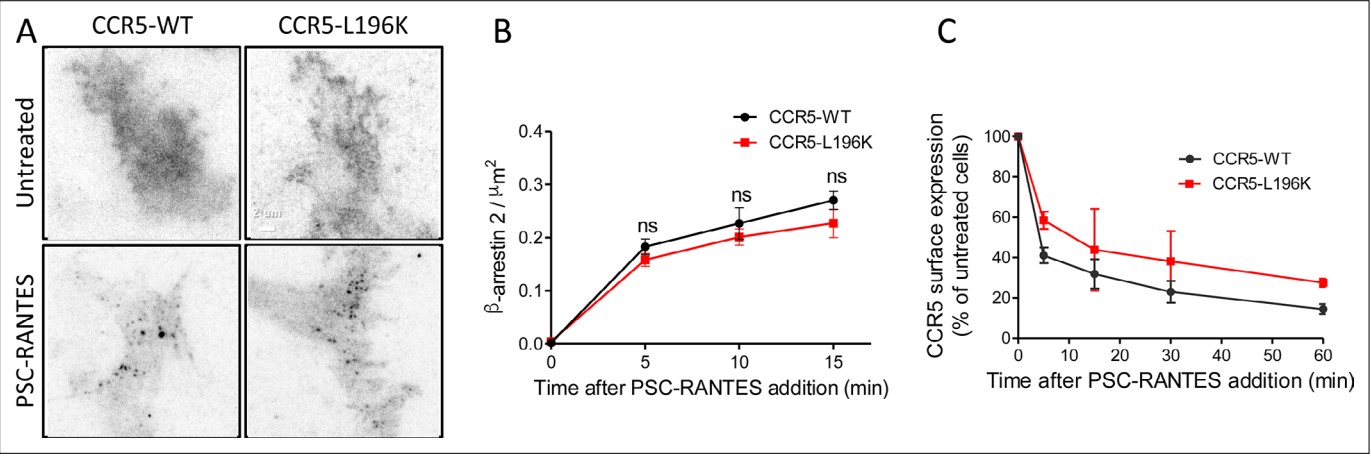

**Figure 6.** Dimerization through TM5 unaffects β-arrestin 2 recruitment to CCR5 but alters its trafficking. (**A, B**) TIRF microscopy on FLAG-ST-CCR5-WT and FLAG-ST-CCR5-L196K cells expressing βarr2-GFP. Cells were stained with M2-Cy3 for FLAG detection and treated or not with 3 nM PSC-RANTES for the indicated times. (**A**) βarr2-GFP spots were detected on TIRF images from untreated cells or cells treated 10 min with PSC-RANTES. Scale bar 2 μm. (**B**) Quantification of the βarr2-GFP spots detected over time using ICY software and spot detector plugin (mean +/-SEM, n=at least 6 cells), Unpaired t test: p≥0.05, ns. (**C**) CCR5 internalization. Cell surface expression of FLAG-ST-CCR5-WT or FLAG-ST-CCR5-L196K was monitored by flow cytometry in stable HEK 293 cell clones after stimulation with a saturating concentration of PSC-RANTES (20 nM) for the indicated time. The percentage of total bound anti-FLAG antibody was calculated from the mean fluorescence intensity relative to untreated cells (mean ± SD from two independent experiments).

The online version of this article includes the following source data and figure supplement(s) for figure 6:

**Source data 1.** Source data for *Figure 6*.

**Figure supplement 1.** β−arrestin 2 recruitment to CCR5 upon PSC-RANTES stimulation.

**Figure supplement 1—source data 1.** Source data for *Figure 6—figure supplement 1*.

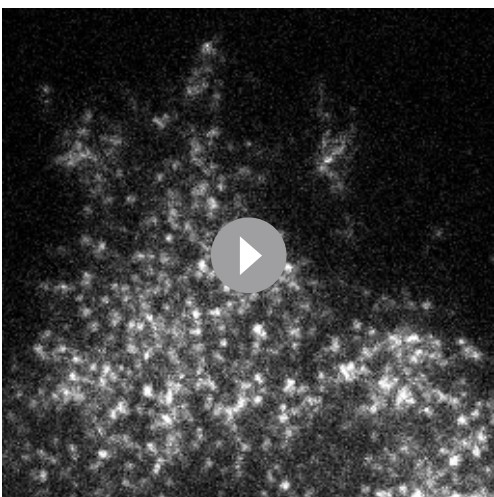

**Video 7.** TIRF movie acquired at 30 Hz of a cell stably expressing eGFP-CCR5-L196K and treated by PSC-RANTES (20 nM) for 2 min.

https://elifesciences.org/articles/76281/figures#video7

cells compared to CCR5-WT expressing cells (*Figure 6—figure supplement 1*). We interpreted this as a consequence of the higher density of receptors per spot for CCR5-WT favoring the probability of βarr2 to colocalize with the receptor in our conditions. These results indicated that the lack of immobilization and clustering of activated CCR5-L196K (*Figure 5E and F*) is not due to a default of βarr2 recruitment.

We next evaluated PSC-RANTES-induced internalization of the dimerization-compromised mutant compared to the WT receptor in feeding experiments using FLAG-ST-CCR5 expressing cells (*Delhaye et al., 2007*; *Jin et al., 2018*). A saturating concentration of PSC-RANTES decreased cell surface expression of both receptors, but not in the same proportion (*Figure 6C*), suggesting that CCR5 dimerization impacted its internalization process. These results supported that dimerization regulated activated receptor mobility and internalization. Note that, while dimerization is a pre-requisite to the immobilization of the receptor, it was not essential for receptor internalization. This suggests that receptor massive immobilization is not an absolute requirement for receptor internalization.

## Distinct HIV-1 envelope glycoproteins gp120 differently influenced CCR5 dynamics

Pharmacological studies suggested that distinct CCR5 conformations at the cell surface differentially engaged distinct HIV-1 envelope glycoproteins gp120 (*Colin et al., 2018*). Since we showed here that CCR5 mobility and ligand engagement are intrinsically linked, we used our mobility classification method to characterize the effect of different HIV-1 gp120s on CCR5 mobility and tested in living cells whether different gp120s engaged different conformational states of CCR5.

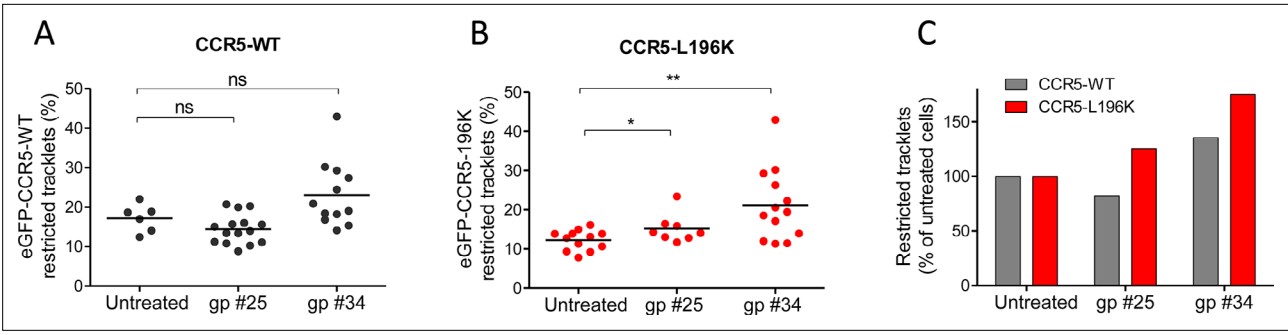

**Figure 7.** HIV-1 gp120s binding restricts eGFP-CCR5 mobility. Soluble gp120s were incubated 30 min at RT in the presence of soluble CD4 (ratio sCD4/gp120>5) to allow their binding to CCR5. Then, gp120-sCD4 complexes were added to live eGFP-CCR5-WT or eGFP-CCR5-L196K expressing cells during at least 20 min before single particle analysis. The proportion of restricted tracklets after gp #25 and gp #34 treatment (100 nM) (in complex with sCD4) on eGFP-CCR5-WT (**A, C**) or eGFP-CCR5-L196K (**B, C**) expressing cells was represented (n=3 independent experiments). Unpaired t test: **p≤*0.005*; ***p≤*0.0001*; ns p≥*0.05*.

The online version of this article includes the following source data and figure supplement(s) for figure 7:

**Source data 1.** Source data for *Figure 7*.

**Figure supplement 1.** HIV-1 gp120s binding restricts FLAG-ST-CCR5 mobility.

**Figure supplement 1—source data 1.** Source data for *Figure 7—figure supplement 1*.

We tested the effect of two soluble gp120s, gp #25 and gp #34, described to induce distinct conformational rearrangements in CCR5 (*Jacquemard et al., 2021*), and to have different binding capacities to the receptor and fusogenic efficacies (*Colin et al., 2018*). Twenty min of gp120 exposure slightly modulated the mobility of eGFP-CCR5-WT (and FLAG-ST-CCR5-WT), although this trend was not statistically significant (*Figure 7A and C*) (*Figure 7—figure supplement 1*). However, and in contrast to what we observed using chemokines as ligands, the HIV-1 gp120s immobilized eGFP-CCR5-L196K, with gp #34 having the highest effect (*Figure 7B and C*). This suggested (i) that gp120s stabilized CCR5 conformations, which were different from those stabilized by chemokines, and (ii) that different envelopes also stabilized differently CCR5 conformations, in accordance with our previous result (*Colin et al., 2013*; *Colin et al., 2018*).

## Discussion

In this study, we developed a statistical method to classify the motion of fluorescent particles at the cell surface. We applied this method to track eGFP-CCR5 or anti-FLAG Cy3 bound CCR5 under different stimuli and different conformations. We obtained the same results with the two models supporting that our findings are independent of the model used. We showed that the receptor fluctuates between Brownian and restricted motions at the cell surface, depending on (1) precoupling to Gi proteins at the basal state; (2) the type of ligand bound to the receptor, and in particular its efficacy on receptor activation and interaction with β-arrestins; and (3) receptor dimerization. Indeed, CCR5 mobility restriction following agonist stimulation were dependent on β-arrestins recruitment and receptor dimerization, but were independent of receptor interaction with Gi proteins. This study demonstrated that coupling receptor motion tracking to a statistical classification of trajectories is a powerful approach to characterize the dynamic behaviors of functionally different receptor populations at the plasma membrane.

### Diversity of ligand-free forms of CCR5 at the cell surface

Quantitative analysis of the motion of CCR5 particles and their composition within the fluorescent spots present at the cell membrane of HEK 293 cells revealed in the basal state (i) two classes of receptor trajectories, Brownian and restricted (*Figure 2*) and (ii) different oligomeric states (*Figure 5*) with low (50 %), medium (40 %), and high fluorescence intensity (10 %). These features shared with other GPCRs (*Gormal et al., 2020*; *Martínez-Muñoz et al., 2018*; *Sungkaworn et al., 2017*; *Tabor et al., 2016*; *Veya et al., 2015*), established the existence of multiple CCR5 forms at the cell membrane.

In addition, our statistical method highlighted a fluctuation between Brownian and restricted states during the same trajectory, suggesting the existence of transient populations of receptors (*Figure 2B*). The change in mobility between periods of confinement separated by free diffusion could be attributed to the molecular organization of the receptor oscillating between different oligomeric forms at the cell surface (monomers, dimers, oligomers), as proposed for CCR5 (*Jin et al., 2018*) or other receptors (*Möller et al., 2020*; *Kasai et al., 2018*; *Martínez-Muñoz et al., 2018*; *Tabor et al., 2016*). In agreement with this, we observed differences in mobility between high and low order oligomeric forms of CCR5 (*Figure 5C*). Change in mobility could also be linked to a transient association of the receptor with the cytoskeleton regardless of its oligomeric status (*Calebiro and Koszegi, 2019*) and/or to transient coupling to G proteins, leading to a transient immobility of the receptor in the basal state. This latter hypothesis is supported by our data in the presence of PTX (*Figure 4A*) or in the presence of the inverse agonist MVC (*Figure 3A and B*), which both uncouple the receptor from G proteins and decreased the proportion of immobile receptors. These data are consistent with dual-color TIRF-M analysis of adrenergic receptor and G protein, showing that an active receptor-G protein complex is formed in a confined region of the plasma membrane at the basal state and lasts around 1 s (*Sungkaworn et al., 2017*). However, they contrast with a study on mGluR3 showing higher mobility of the receptor when complexed with G protein (*Yanagawa et al., 2018*). This suggested that dynamics of distinct GPCRs can be differently impacted by coupling to G proteins. Regarding β-arrestin association, we showed using siRNA that CCR5 was not precoupled to β-arrestins in its basal state (*Figure 4C*). This result suggests that CCR5 conformations, which bind to G proteins are not recognized by β-arrestins. This is consistent with the idea that the conformations of receptors interacting with G proteins and β-arrestins are different (*Lagane et al., 2005*).

## Different ligands recognize/stabilize different sets of CCR5

We showed that CCR5 mobility is influenced differently according to the ligand it binds. Chemokine-induced activation of eGFP-CCR5-WT (or FLAG-ST-CCR5-WT) decreased receptor mobility and leads to clustering (*Figure 3B and D*), effects not observed with the inverse agonist MVC and abolished by MVC (*Figure 3A and B* and *Figure 3E and F*). This result reinforces the link between GPCR mobility and ligand binding proposed for GPCRs of different classes (*Gormal et al., 2020*; *Möller et al., 2020*; *Veya et al., 2015*; *Yanagawa et al., 2018*).

We also showed that two agonists with different efficacies, and targeting different subsets of receptors (CCL4 and PSC-RANTES) (*Escola et al., 2010*; *Jin et al., 2018*), restricted receptor motion in a different proportion (*Figure 3*). Therefore, characterizing ligands by their impact on receptor motion opens a new way to classify biased ligands.

Applied to viral envelope glycoproteins, our tracking approach revealed that HIV-1 gp120s displayed an agonist-like influence on CCR5 mobility, albeit to different extent according to the nature of the gp120 (*Figure 7*). This feature contrasts with the cryo-EM structure of the CD4-gp120-CCR5 complex, showing that CCR5 adopts inactive confomation (*Shaik et al., 2019*). However, it is in line with gp120s-induced CCR5 signaling (*Brelot and Chakrabarti, 2018*; *Flanagan, 2014*) and with recent MD simulations showing that gp120 binding reorients charactteric microswitches involved in GPCR activation (*Jacquemard et al., 2021*). The fact that the fraction of immobilized receptors varied between gp120s could reflect that they do not bind to/stabilize the same CCR5 conformations, as previously shown (*Colin et al., 2018*; *Jacquemard et al., 2021*), and suggests that these gp120s behave themselves as biased agonists. These features of gp120s will help understand the determinants of HIV-1 tropism.

## Receptor motion tracking analysis revealed that dimerization regulates the fate of activated CCR5

Our results suggest that receptor dimerization may regulate precoupling of CCR5 to Gi proteins. Indeed, the mobility of the dimerization-compromized mutant eGFP-CCR5-L196K was not affected by PTX treatment (*Figure 5D*), in contrast to the WT receptor (*Figure 4A*). This suggests that most eGFP-CCR5-L196K receptors that reside preferentially as monomers are not coupled to Gi proteins in the basal state, in agreement with previous conclusion on CXCR4 (*Möller et al., 2020*). Alternatively, but not exclusively, CCR5-L196K dimers might also be impaired in their ability to be precoupled to Gi proteins, contrary to WT receptor dimers.

Our analysis suggests that dimerization is a pre-requisite to receptor immobilization and clustering upon activation by chemokine agonists. Indeed, unlike eGFP-CCR5-WT, eGFP-CCR5-L196K receptors are only marginally immobilized in the presence of PSC-RANTES (*Figure 5E*). This result is not due to impaired binding of the chemokine, because we controlled that PSC-RANTES induced efficient ERK1/2 activation (*Figure 5—figure supplement 1*) and endocytosis of the mutant receptor (*Figure 6C*). Receptor immobility and clustering were independent of Gi protein coupling, as exemplified by unaffected CCR5 mobility after 10 min of agonist stimulation in PTX pre-treated cells (*Figure 4B*), but most likely related to uncoupled and desensitized form of CCR5 that accumulate in CCS (clathrin-coated structures), as proposed (*Grove et al., 2014*; *Yanagawa et al., 2018*). This hypothesis was strengthened with the essential role of β-arrestins in activated receptor immobility and clustering (*Figure 4D*; *Markova et al., 2021*) and with studies showing that β-arrestins recruitment depends on the efficiency of ligand to trigger CCR5 internalization (*Jin et al., 2018*; *Tarancón Díez et al., 2014*; *Truan et al., 2013*). We cannot rule out that activated receptor clustering may in addition correspond to an accumulation of receptor in early endosome for a second phase of activation (*Irannejad et al., 2013*).

In line with this, we showed that dimerization regulates endocytosis (*Figure 6C*). The lack of immobilization of the dimerization-compromised mutant leads to a suboptimal internalization of the receptor. This is not due to a default in βarr2 recruitment since CCR5-WT and CCR5-L196K similarly recruited βarr2-GFP to the plasma membrane (*Figure 6A and B*). Effective interaction of βarr2 with CCR5-L196K, which is mostly monomeric in the basal state, is consistent with structural studies showing GPCR-arrestin complexes in a 1:1 arrangement (*Kang et al., 2015*). We propose a model in which receptor oligomerization might be an essential requirement for β-arrestins to trigger receptor clustering and immobilization. A concerted self-association of arrestins may favor this process (*Kim et al., 2011*). Indeed, PSC-RANTES induces strong β-arrestins clustering (*Tarancón Díez et al., 2014*;

*Truan et al., 2013*). We speculate that the co-clustering of β−arrestins with receptors may serve as a platform helping to concentrate cargo for optimal and productive internalization. Note that, while dimerization is a pre-requisite for receptor immobilization (*Figure 5*), it is not essential for receptor internalization (*Figure 6C*).

Differential effects of gp120 on immobilization of CCR5-WT and CCR5-L196K (*Figure 7*), compared to chemokines (*Figure 5*), could also be explained by differences in β-arrestins ability to cluster dimers, linked to differences in the stabilized conformations of receptors.

Finally, our study suggested that CCR5 can be activated whether monomeric or dimeric. We showed that eGFP-CCR5-L196K, while mostly monomeric in its basal state (*Figure 5B*), is able to activate ERK1/2 (*Figure 5—figure supplement 1*) and is still internalized after stimulation (*Jin et al., 2018*; *Figure 6C*). This is consistent with studies reporting that GPCR monomers can be active enough on their own to be functional (*Whorton et al., 2007*).

In summary, our receptor motion tracking analysis established that a diversity of CCR5 forms exists at the surface of living cells and that distinct ligands stabilize different receptors. This approach also revealed that receptor dimerization is involved in Gi protein-coupling in the basal state, and in the ability of β arrestin 2 to cluster receptors, therefore impacting the mobility of activated receptors. These findings, point out that receptor conformation regulates GPCRs signaling and fate after activation. In addition, our work suggested that the different receptor conformations likely engaged different ways of regulation, expanding GPCRs functions.

# Materials and methods

**Key resources table**

| Reagent type (species) or resource | Designation | Source or reference | Identifiers | Additional information |
|---|---|---|---|---|
| Cell line (*Homo sapiens*) | HEK293 cells | ATCC | CRL-1573; RRID: CVCL_0045 | Human embryonic kidney (female) |
| Cell line (*Homo sapiens*) | A3.01-R5 | *Colin et al., 2013* | | CEM T cell line derivated cells |
| Antibody | α-GFP (mouse monoclonal) | Roche | 11814460001 | Flow cytometry dilution (1: 100) |
| Antibody | α-CCR5 2D7 (mouse monoclonal) | BD-Biosciences | 555,991 | Flow cytometry dilution (1: 500) |
| Antibody | FLAG tag M2 (mouse monoclonal) | Sigma | Cat# F3165 | Flow cytometry dilution (1: 750) |
| Antibody | FLAG tag M2-Cy3 (mouse monoclonal) | Sigma | Cat# A9594 | TIRF microscopy dilution (1: 1000) |
| Antibody | Phospho ERK1/2 (mouse monoclonal) | Cell signaling | Cat# 9,106 | Western blot dilution (1:2500) |
| Antibody | ERK2 (Rabbit polyclonal) | Santa-Cruz Biotech | Cat# sc-154 | Western blot dilution (1:750) |
| Antibody | Goat anti-mouse HRP (rat monoclonal) | BD-Biosciences | Cat# 559,751 | Western blot dilution (1:120000) |
| Antibody | Goat anti-rabbit HRP (goat polyclonal) | Jackson | Cat# 111-035-144 | Western blot dilution (1:3500) |
| Antibody | Goat anti-mouse phycoerythrin (PE) (goat polyclonal) | BD-Biosciences | Cat# 550,589 | Flow cytometry dilution (1:100) |
| Recombinant DNA reagent | pmCherry- (plasmid) | other | | Provided by F. Perez (Institut Curie). |
| Recombinant DNA reagent | peGFP-CCR5 (plasmid) | other | | Provided by F. Perez (Institut Curie). |
| Recombinant DNA reagent | peGFP-CCR5-L196K (plasmid) | This paper | | Contains a point mutation in CCR5 at position L196. |
| Recombinant DNA reagent | pFLAG-SNAP-CCR5-WT (plasmid) | *Jin et al., 2018* | | Provided by Cisbio |

*Continued on next page*

*Continued*

| Reagent type (species) or resource | Designation | Source or reference | Identifiers | Additional information |
|---|---|---|---|---|
| Recombinant DNA reagent | pFLAG-SNAP-CCR5-L196K (plasmid) | *Jin et al., 2018* | | Introduction of a lysine in position L196 |
| Recombinant DNA reagent | pβarr2-GFP | *Storez et al., 2005* | | Provided by S. Marullo (Institut Cochin) |
| SiRNA reagent | βarr1/2 (siRNA) | Dharmacon | | See Materials and methods for sequence |
| SiRNA reagent | Scrambled (siRNA) | Dharmacon | | See Materials and methods for sequence |
| Soluble protein | HIV-1 gp120 #25, #34 | *Colin et al., 2018* | | Gp120 from PBMCs of patients in early or late HIV-1 infection stage. See details in 'cell culture and reagents' section of 'Materials and methods' |
| Soluble protein | Human sCD4 | *Colin et al., 2018* | | See details in 'cell culture and reagents' section of 'Materials and methods' |
| Chemical compound, drug | Maraviroc | NIH | Cat# ARP-11580 | CCR5 inverse agonist |
| Chemical compound, chemokine | CCL4 | This paper | | Provided by F. Baleux (Institut Pasteur) |
| Chemical compound, drug | PSC-RANTES | NIBSC | Cat# ARP973 | CCR5 agonist |
| Chemical compound, chemokine | SDF-1 | Peprotec | Cat# 300–28 A | CXCR4 agonist |
| Chemical compound, drug | Pertussis Toxin | Sigma | Cat#179 A | 100 ng/ml |
| Software, algorithm | Prism | GraphPad | 8.1.1 | |
| Software, algorithm | ICY | Open access | Version 2.4.0.0 | https://icy.bioimageanalysis.org/ |
| Software, algorithm | MATLAB | MathWorks | R2017a | |

## Cell culture and reagents

The HEK 293 cells stably expressing FLAG-SNAP tagged- CCR5-WT (FLAG-ST-CCR5-WT) and FLAG-SNAP tagged-L196K (FLAG-ST-CCR5-L196K) and the A3.01 human T cell line stably expressing CCR5 (A3.01-R5) were previously described (*Colin et al., 2013*; *Jin et al., 2018*). These cell lines were maintained in Dubelcco's modified Eagle medium (DMEM) (Thermo Fisher Scientific) or RPMI 1640 medium supplemented with 10% Fetal Bovine Serum (FBS, GE Healthcare) and 100 µg/ml penicillin/streptomycin (Life technologies).

The CCR5 inverse agonist maraviroc (MVC) was obtained from the National Institutes of Health. The native chemokine CCL4 was chemically synthetized by F. Baleux (Institut Pasteur, Paris, France). The chemokine analog PSC-RANTES (N-α-(n-nonanoyl)-des-Ser(1)-[L-thioprolyl(2), L cyclohexyl-glycyl(3)] RANTES(4-68)) was obtained through the Center for Aids reagents, National Institute for Biological Standards and Control (NIBSC, UK). The primary antibodies used are the anti-GFP (Roche), the anti-CCR5 2D7 mAb (BD-Biosciences); the anti-FLAG monoclonal antibodies M1 or M2 or M2-Cy3 (Sigma-Aldrich), the phospho-ERK ½ (Cell Signaling) and ERK2 (Santa Cruz). Secondary antibodies used were a phycoerythrin (PE)-conjugated anti-mouse antibody (BD Biosciences), a horseradish peroxidase (HRP)-conjugated anti-mouse antibody (BD Pharmingen) and a horseradish peroxidase (HRP)-conjugated anti-rabbit antibody (Jackson). The toxin from Bordetella pertussis (PTX) used at a 100 ng/ml concentration were from Sigma. The βarr1/2 siRNA (5'-ACCUGCGCC UUCCGCUAUG-3') and a scrambled siRNA (control, 5'-UGGUUUACAUGUCGACUAA-3') (Dharmacon) were transfected by RNAimax (Invitrogen) according to the instructions of the manufacturer, as described (*Jin et al., 2014*). To select siRNA positive cells, cells were co-transfected with a plasmid coding the fluorescent protein mcherry (gift of F. Perez, Institut Curie). The construct encoding for GFP fusion of wild-type β-arrestin 2 (βarr2-GFP) have been described previously (gift of S. Marullo) (*Storez et al., 2005*). Soluble, monomeric HIV-1 glycoprotein gp120 was produced using a semliki forest virus (SFV) system as described (*Benureau et al., 2016*; *Colin et al., 2018*). The sequence coding for gp120 #25 and gp120 #34 were from PBMCs of patients collected early after seroconversion or in the AIDS stage of infection, respectively (*Colin et al., 2018*). Recombinant

soluble CD4 (sCD4), produced in S2 cell lines, was purified on a strep-Tactin column using the One-STrEP-tag fused to the CD4 C-tail as a bait (production and purification of recombinant proteins technological platform, C2RT, Institut Pasteur).

## Generation of cell lines

The eGFP-CCR5 plasmid was a gift of F. Perez (Institut Curie, Paris, France). eGFP-CCR5 was expressed from the CMV promoter. The mutant eGFP-CCR5-L196K (substitution of L196 with a lysine) was generated by site-directed mutagenesis using the QuickChange II Mutageneis kit (Agilent Technologies) according to the manufacturer's instruction. This mutant was verified by sequencing (Eurofins). HEK 293 cells stably expressing eGFP-CCR5-WT and HEK 293 cells stably expressing eGFP-CCR5-L196K were generated by calcium phosphate transfection and cultured for several weeks in 1 mg/ml G418 (Geneticin, Invitrogen). Cell clones were screened and sorted by flow cytometry (Attune NxT flow cytometer, Thermo Fisher) using an anti-GFP monoclonal antibody.

## Receptor cell surface expression levels and internalization measured by flow cytometry

Flow cytometry was used to quantitate the internalization of FLAG-ST-CCR5-WT compared to FLAG-ST-CCR5-L196K stably expressed in HEK 293 cells (*Delhaye et al., 2007*; *Jin et al., 2018*). We measured the levels of cell surface CCR5 stained with the anti-FLAG M2 antibody and with an anti-mouse coupled to phycoerythrin (PE) after chemokine treatment or not. Cells were incubated with a saturable amount of M2 for 45 min to label receptors present at the plasma membrane, then incubated in the presence (or not) of 20 nM PSC-RANTES for the indicated time at 37 °C. Cells were chilled to 4 °C and stained with a PE conjugated anti-mouse IgG. Mean values were used to compute the proportion of internalized receptors as indicated by a decrease of immune-reactive surface with PSC-RANTES compared with untreated cells. Cells were analyzed with Attune NxT flow cytometer (Thermo Fisher). At least 5000 cells were analyzed per experiment using Kaluza software. Background was subtracted using the fluorescence intensity obtained on the parental HEK 293 cells.

## Chemotaxis

CCR5 expressing A3.01 cells (A3.01-R5, $1.5 \times 10^5$), pre-treated or not with PTX (100 ng/ml) during 3 hr, in prewarmed RPMI-1640 supplemented with 20 mM Hepes and 1% serum, were added to the upper chambers of HTS-Transwell-96 Well Permeable Supports with polycarbonate membrane of 5 µm pore size (Corning). PSC-RANTES (33.7 nM) or SDF-1 (control, 10 nM) was added to the lower chambers. Chemotaxis proceeded for 4 hr at 37 °C in humidified air with 5% $CO_2$. The number of cells migrating across the polycarbonate membrane was assessed by flow cytometry with Attune NxT flow cytometer (Thermo Fisher). Specific migration was calculated by subtracting spontaneous migration from the number of cells that migrated toward the chemokine.

## Phospho-ERK1/2 measurements

FLAG-ST-CCR5 expressing cells ($1.5 \times 10^5$) were grown in 24-well plates pretreated with poly-D-lysine and rendered quiescent by serum starvation for 16 hr prior to incubation with or without CCL4, as indicated. Plates were placed on ice and the cells were then scraped into lysis buffer composed of 0.5% n-dodecyl-β-D-maltoside (NDM), 0.2% iodoacetamide, protease and phosphatase inhibitors in mTBS. After 30 min, samples were centrifuged and heated for 10 min at 60 °C before resolution of equal amounts of proteins on SDS-PAGE. The proteins were transferred to nitrocellulose membranes, and immunoblotting were carried out using the indicated antibodies. Immunoreactivity was revealed using a secondary antibody coupled to HRP. Band intensities on the same film were quantified by densitometry.

## βarrestin 2 recruitment at the plasma membrane

FLAG-ST-CCR5 expressing cells, transfected with βarr2-GFP, were plated on MatTek plates 72 hr before imaging. Cells were stained with the anti-FLAG M2-Cy3 (5 min) and incubated in the presence or absence of 3 nM PSC-RANTES in DMEM/1%BSA medium for the indicated time. Cells were put on ice and fixed with paraformaldehyde (PFA) 4% at 4 °C for 40 min before three washes in PBS. Experiments were performed using a Elyra 7 microscope (Carl Zeiss Gmbh) equipped with two sCMOS

cameras PCO Edge 4.2, and using an alpha Plan Apo 63 x/1.46 oil objective, a 488 nm (500 mW) and a 561 nm (500 mW) laser line, and a quad band filter coupled to BP 495–550 or BP 570–620 filters. All TIRF images analyses were performed using ICY software and the spot detector and the colocalization studio plugins. The number of spot detected per cell was normalized to the size of the cell surface.

### Live cell TIRF imaging

Round 25 mm No. 01 glass coverslips (Fisher Scientific) were pre-cleaned with 70% ethanol followed by acetone, with three consecutive washes in ddH2O. $1.15 \times 10^5$ cells were plated onto pre-cleaned coverslips 72 hr before imaging. Cells were imaged in TIRF medium (25 mM HEPES, 135 mM NaCl, 5 mM KCl, 1.8 mM CaCl2, 0.4 mM MgCl2, 4.5 g/l glucose and 0.5% BSA, pH 7.4). For eGFP tracking, movies were acquired with an LSM 780 Elyra PS.1 TIRF microscope (Zeiss) equipped with an EMCCD Andor Ixon 887 1 K camera, and using an alpha Pin Apo 100 x/1.46 oil objective, a 488 nm (100 mW) HR solid laser line, and a BP 495–575+LP 750 filter to detect eGFP-CCR5. Image acquisition was done at 1 frame / 33 ms (30 Hz) (100–200 frames), with an illumination intensity <0.38 kW/cm² (tracking) or 0.7 kW/cm² (fluorescence intensity) at 37 °C. Under these conditions, the intensity of the spots is stable throughout the duration of the acquisition. Approximately 5–10 cells were acquired per condition, per experiment. For FLAG-ST-CCR5 tracking, movies were acquired with a TIRF microscope (IX81F-3, Olympus) equipped with a X 100 numerical aperture 1.45 Plan Apo TIRFM Objective (Olympus) and fully controlled by CellM (Olympus). Images were collected using an IxonEM camera (DU885, Andor). Image acquisition was done at 10 Hz with an illumination intensity of about 0.1 kW/cm².

All live-imaging movies were analyzed using the open-source software Icy (Institut Pasteur).

### Track analysis protocol

#### Tracking receptors in TIRF imaging with Icy software

To automatically detect eGFP-CCR5 tracks at the plasma membrane upon time, we used the software Icy (http://icy.bioimageanalysis.org) and the plugin *Spot tracking*, which reports their *xy* displacement and intensities, as previously described in *Bertot et al., 2018*. *Spot tracking* was set to detect spots with approximately 3 pixels, and a threshold of 135. All other parameters were as default. Tracks were analyzed with the *Track manager* plugin. All data was exported to *Excel* for further analysis.

Tracks containing more than 10% of virtual detections and more than three successive virtual detections were excluded from the track classification.

#### Splitting tracks into tracklets

We deal with trajectories that have very different lengths and we want to estimate motion variations along the trajectory. Thus, we split all long tracks into several tracklets in order to better classify local motions. According to Section 1, this is done by setting N = 5 and considering only the tracks with length larger than 6. Then, the different successive tracklets are defined by using the position between the $(5k)^{th}$ and $(5(k+1))^{th}$ frame with $k \geq 0$.

#### Detecting immobile receptors

To classify tracklets and identify distinct receptor dynamics, we first identified immobile receptors. In time lapse imaging, a tracklet X is defined by the vector of its successive positions at the different time frames $X = (X_0, \ldots, X_{N-1})$, with *N* the length of the tracklet. We considered that a receptor was immobile if.

$$\max_{i \neq j=0,\ldots,N-1} \|X_i - X_j\| < \sqrt{2}\, l$$

where *l* is the size of the object (*l=2* pixels typically). In other words, the previous criterion states that a tracklet is immobile if the maximal distance between two different positions is at most equal to the length of the diagonal of the square of edge *l*.

#### The three types of motion of mobile receptors

To classify the other tracklets corresponding to mobile receptors, we used the statistical method introduced in *Briane et al., 2018*, which allows to distinguish three main types of motions:

(i) **Brownian motion**: the object (receptor) evolves freely and its trajectory is denoted by $\sigma B_t$ where $\sigma$ is called the *diffusion coefficient*. The position of the object $X_t$ at time $t$ is given by $X_t = X_0 + \sigma B_t$. Brownian increments $\sigma dB_t$ at each time are independent and normally distributed.

(ii) **Directed motion**: the object is actively transported by a deterministic force, and its motion can be modelled by the following stochastic differential equation:

$$dX_t = \mu dt + \sigma dB_t,$$

where µ is a 2D-vector called *drift* and represents the deterministic force, and σ is the *diffusion coefficient* modelling the random Brownian motion.

(iii) **Confined motion**: the object is confined in a domain or evolves in an open but crowded area. This kind of motion can be modeled by an Ornestein-Uhlenbeck process:

$$dX_t = -\lambda \left( X_t - \mu \right) dt + \sigma dB_t.$$

We refer to *Durrett, 2018* for more properties about Brownian motion and stochastic calculus.

## Statistical classification of mobile tracklets

The motion classification criterion defined in *Briane et al., 2018* essentially considers the ratio between the maximal distance from the initial point and the length of the tracklets. This can be evaluated by defining the following statistics.

$$S\left( X, N \right) = \frac{\max\limits_{i=0,\dots,N} \left| X_{t_i} - X_{t_0} \right|}{\left[ \frac{1}{2} \sum\limits_{i=1}^{N} \left| X_{t_i} - X_{t_{i-1}} \right|^2 \right]^{\frac{1}{2}}}$$

where |.| denotes the 2D-Euclidean norm. The classification is made by using the quantiles of order $\alpha$ and 1-$\alpha$ ($\alpha = 0.05$) of such a statistic for Brownian tracklets.

These quantiles, denoted by q($\alpha$) and q(1-$\alpha$) respectively, depend on $\alpha$ and N, and can be computed by Monte Carlo simulations (see *Briane et al., 2018*). This essentially consists in simulating a high number of Brownian tracklets, computing their statistics values and then evaluating the quantiles.

Then the tracklet motion is said to be confined if S(X,N)<q($\alpha$), directed if S(X,N)>q(1-$\alpha$), and Brownian otherwise. For N=5 and $\alpha = 0.05$, we obtained q($\alpha$)=0.724 and $q(1 - \alpha) = 2.464$.

## From local classification of tracklet motion to global analysis of receptors' tracks

The above statistical classifier allows estimating the local motion of each receptor. In a second time, we analyzed the difference of tracklet motions along the same longer receptor track. In particular, we evaluated if a receptor changed its type of motion along its trajectory.

Finally, our statistical framework for classifying tracklets motion provided a two-scales picture of the receptors' dynamic behavior: the classification of tracklets provided a global estimation of receptors' motion, while the identified changes of receptors' motion along their full trajectories indicated the stability of each receptor's motion.

## Simulating synthetic receptors' trajectories

To evaluate the robustness and accuracy of tracklet classification, we first simulated in Matlab n=*100* confined trajectories ($dX_t = -\lambda \left( X_t - \mu \right) dt + \sigma dB_t$) with length N+*1*, where N=5 or 10 is the length of used tracklets for classification. Diffusion coefficient $\sigma$ was fixed to $\sigma = \sqrt{2}$ and we varied the confinement parameter $\lambda$ from $\lambda = 0$ to $\lambda = 6$ (step =0.2). We then measured the accuracy of classification with $p_N\left(0\right) = \frac{\{\# \textit{ tracklets classified as confined}\}}{\{\# \textit{ simulated tracklets } (n)\}}$. As expected, the classification accuracy increases with tracklet length *N* (*Figure 1—figure supplement 1*). To account for the risk of mistracking or a change in receptor dynamics, that also increases with tracklet length, we then modeled a generic *perturbation* in receptor tracking with a standard exponential distribution with rate $\rho$. Therefore, the conditional probability $p_N\left(\rho\right)$ for a confined tracklet with length N to be correctly classified is given by $p_N\left(\rho\right) = p_N\left(0\right) \exp\left(-\rho N\right)$ where $p_N\left(0\right)$ is the classification accuracy when no risk of mistracking or dynamic change is considered.

In a second time, to measure the robustness of classification to image noise and particle (receptor) density, we simulated a mixture of n=*1,000* of Brownian and Confined trajectories (the percentage of simulated confined trajectories was fixed to 10, 50, or 100%) and generated the associated synthetic fluorescence time-lapse sequences using a mixed Poisson-Gaussian model as described in *Lagache et al., 2021*. We implemented the simulator of synthetic tracklets of fluorescent spots in ICY (Plugin *Dynamics Simulator*). Using our simulator, we varied the signal-to-noise ratio (SNR) from 10 to 2 (we measured a mean SNR~10 in our experimental dataset). Concerning the density of receptors' spots, we varied the size of the simulated sequence from xy = 1600 × 1600 pixels to 800 × 800 pixels and 400 × 400 pixels, corresponding respectively to spots' density = 0.039, 0.16, and 0.63 $spots/\mu m^2$ , the measured density being <0.5 $spots/\mu m^2$ in most experiments.

## Stoichiometry analysis

Icy software was used to determine the intensity distribution of eGFP-spots. Spots were detected using the Spot detector wavelet-based algorithm (*Olivo-Marin, 2002*), and then converted to ROIs with 2 pixels radius. Data was exported to Excel. We observed a multimodal distribution of eGFP spots' intensities, and we decided to use the AIC criterion (Akaike information criterion; *Akaike, 1974*) to uncover the number of modes in intensity distribution. Each mode putatively corresponds to a number of molecules. Therefore, statistical characterization of the multimodal distribution of eGFP spots' intensity will help to classify each spot with respect to its mode and, therefore, to its estimated number of molecules.

AIC analysis starts with the modeling of the empirical distribution e(x) of eGFP spots'intensities with a weighted sum of Gaussian laws,

$$e\left(x\right) = \sum_{i=1}^{p} \alpha_i N\left(\mu_i, \sigma_i\right)$$

where p is the number of Gaussian laws in the mixture, $\alpha_i$ the weight of each law and $(\mu_i, \sigma_i)$ the corresponding mean and variance. For a fixed p, we first searched for the optimal parameters $(\alpha_i^*, \mu_i^*, \sigma_i^*)$, for $i = 1..p$ that maximize the likelihood L of the model to the data:

$$L_p\left(\alpha_1, \mu_1, \sigma_1, \ldots, \alpha_p, \mu_p, \sigma_p\right) = \prod_{j=1}^{n}\left[\sum_{i=1}^{p} \frac{\alpha_i}{\sqrt{2\pi\sigma_i}} \exp\left(-\frac{(x_j-\mu_i)^2}{2\sigma_i}\right)\right]$$

where $(x_1, x_2, \ldots, x_n)$ are the observed eGFP intensities in the considered frame of the time-lapse sequence.

This first step of the AIC analysis provides the calibrated parameters $\left(\alpha_i^*, \ \mu_i^*, \sigma_i^*\right)_{i=1..p}$ when fitting a *p*-mixture model to data. Then, we computed the optimal number of modes $p^*$ that would describe the different populations of eGFP spots with respect to their estimated number of molecules by minimizing the AIC:

$$AIC\left(p\right) = 2k_p - 2\log\left(L_p^*\right)$$

where $L_p^*$ is the maximized likelihood the p-mixture model, and $k_p = 3p - 1$ is the number of free parameters of the *p*-mixture model.

## Acknowledgements

We are grateful to Françoise Baleux (Institut Pasteur), Stefano Marullo (Institut Cochin) and Franck Perez (Institut Curie) for the gifts of chemokines and plasmids. We acknowledge Oliver Hartley (University of Genova) and the Programme EVA Centre for AIDS Reagents for the chemokine derivative PSC-RANTES. We acknowledge Stéphane Petres from the Production and Purification of Recombinant Proteins (PPRP) platform (C2RT, Institut Pasteur) for sCD4 production. We thank Audrey Salles from the Photonic BioImaging (PBI) platform (Imagopole) of Institut Pasteur for microscope maintenance and technical help. We thank Vannary Meas-Yedid Hardy (Institut Pasteur), Stéphane Dallongeville (BioImage Analysis Unit, Institut Pasteur), the Image Analysis Hub (C2RT, Institut Pasteur), Gael Moneron (Synapse and Circuit Dynamics), and VizionR (Paris) for help with the image and data analysis. This work was supported by grants from Agence National de Recherche sur le SIDA et les

hepatitis virales (ANRS), the French Government's Investissement d'Avenir program, Laboratoire d'excellence "Integrative Biology of Emerging Infectious Diseases' (grant ANR-10-LABX-62-IBEID), INCEPTION (ANR-16-CONV-0005) and GET-REDI (ANR-21-CE44-0030). UTechS PBI is part of the France–BioImaging infrastructure network (FBI) supported by the French National Research Agency (ANR-10-INBS-04; Investments for the Future), and acknowledges support from Institut Pasteur, ANR/FBI, the Région Ile-de-France (program 'Domaine d'Intérêt Majeur-Malinf' and DIM1HEALTH), and the French Government Investissement d'Avenir Programme—Laboratoire d'Excellence 'Integrative Biology of Emerging Infectious Diseases' (ANR-10-LABX-62-IBEID) for the use of ELYRA PS1 LSM780 and ELYRA7 microscopes. FM was the recipient of ANR-10-LABX-62-IBEID fellowship, GN of INCEPTION (ANR-16-CONV-0005) fellowship and PC of an ANRS fellowship.

# Additional information

## Competing interests

The other authors declare that no competing interests exist.

## Funding

| Funder | Grant reference number | Author |
| --- | --- | --- |
| Agence Nationale de la Recherche | ANR-10-LABX-62-IBEID post-doctoral fellowship | Fanny Momboisse |
| Agence Nationale de la Recherche | ANR-16-CONV-0005-INCEPTION Post doctoral fellowship | Giacomo Nardi |
| Agence Nationale de Recherches sur le Sida et les Hépatites Virales | Post-doctoral fellowship | Philippe Colin |
| Agence Nationale de Recherches sur le Sida et les Hépatites Virales | | Olivier Schwartz Bernard Lagane |
| Agence Nationale de la Recherche | ANR-10-LABX-62-IBEID | Olivier Schwartz Fernando Arenzana-Seisdedos Nathalie Sauvonnet Jean-Christophe Olivo-Marin Bernard Lagane Thibault Lagache Anne Brelot |
| Agence Nationale de la Recherche | ANR-10-INBS-04 | Jean-Christophe Olivo-Marin |
| Agence Nationale de Recherches sur le Sida et les Hépatites Virales | ANRS-AP19-1 | Anne Brelot |
| Agence Nationale de la Recherche | ANR-16-CONV-0005-INCEPTION | Jean-Christophe Olivo-Marin |
| Agence Nationale de la Recherche | ANR-21-CE44-0030-GET-REDI | Thibault Lagache Anne Brelot |

The funders had no role in study design, data collection and interpretation, or the decision to submit the work for publication.

## Author contributions

Fanny Momboisse, Conceptualization, Formal analysis, Investigation, Methodology, Validation, Writing - original draft; Giacomo Nardi, Conceptualization, Formal analysis, Methodology, Software, Validation, Writing - original draft; Philippe Colin, Formal analysis, Investigation, Validation; Melanie Hery, Nelia Cordeiro, Simon Blachier, Validation; Olivier Schwartz, Fernando Arenzana-Seisdedos,

Jean-Christophe Olivo-Marin, Funding acquisition, Resources, Supervision; Nathalie Sauvonnet, Conceptualization, Formal analysis, Funding acquisition, Resources, Writing - original draft; Bernard Lagane, Conceptualization, Formal analysis, Funding acquisition, Resources, Supervision, Writing - original draft; Thibault Lagache, Conceptualization, Data curation, Formal analysis, Funding acquisition, Methodology, Software, Writing - original draft; Anne Brelot, Conceptualization, Data curation, Formal analysis, Funding acquisition, Investigation, Methodology, Project administration, Resources, Supervision, Validation, Visualization, Writing - original draft

### Author ORCIDs
Philippe Colin http://orcid.org/0000-0002-5552-8908
Olivier Schwartz http://orcid.org/0000-0002-0729-1475
Nathalie Sauvonnet http://orcid.org/0000-0003-0306-3376
Thibault Lagache http://orcid.org/0000-0002-7033-4677
Anne Brelot http://orcid.org/0000-0002-8095-4909

### Decision letter and Author response
Decision letter https://doi.org/10.7554/eLife.76281.sa1
Author response https://doi.org/10.7554/eLife.76281.sa2

## Additional files

### Supplementary files
• Transparent reporting form

### Data availability
The code used for data analysis in MatLab was provided as Figure 1—source code 1. The numerical data used to generate each figure and figure supplement were provided as source data files.

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
