## [Editor Report]

This manuscript seeks to push the frontiers of live-cell single-molecule imaging by tracking the diffusive movements of CCR5 receptors and CCR5 receptor complexes within the plasma membrane of living cells and how these motional behaviors change with physiological stimuli. The results will be important for researchers working at the interface of cell biology and biophysics on membrane-bound receptors.

---

## [Decision Letter]

**Decision letter after peer review:**

[Editors’ note: the authors submitted for reconsideration following the decision after peer review. What follows is the decision letter after the first round of review.]

Thank you for submitting the paper "Single-molecule imaging reveals distinct effects of ligands on CCR5 dynamics depending on its dimerization status" for consideration by *eLife*. Your article has been reviewed by 3 peer reviewers, including Volker Dötsch as the Reviewing Editor and Reviewer #3, and the evaluation has been overseen by a Senior Editor.

Comments to the Authors:

We are sorry to say that, after consultation with the reviewers, we have decided that this work will not be considered further for publication by *eLife*.

While everyone involved in the discussion of your manuscript was convinced that investigating the dynamic behavior of individual receptors on the cell surface is a very important and very ambitious task, concerns were raised whether the experimental data reported support the conclusions. The consensus was that more experimental data are necessary to fully support these conclusions. As these experiments will very likely take longer than 2 months, it is the policy of *eLife* to reject in such cases the manuscript. More details about the discussed problems you can find in the individual reviews below which will hopefully help you to revise the manuscript for submission to another journal.

*Reviewer #1 (Recommendations for the authors):*

The manuscript from Momboisse et al. provides a single molecule tracking (SMT) investigation of the CCR5 receptor on the surface of living cells.

Taking advantage on the ability of SMT to detect individual trajectories and hence monitor the behaviour of specific receptors over time, the authors approach aims at investigating fluctuations in the dynamic behavior of individual receptors in response to alterations in conformational changes of the receptor, either spontaneous, or upon stimulation with specific agonists or antagonists.

The main general hypothesis of the study, with GPCR-wide relevance, is that receptor dynamics can be correlated to its molecular conformation (active vs inactive, in the coarsest description). The specific hypothesis related to CCR5 is that gp120 (HIV-1 envelope glycoprotein) acts as an agonist, and that receptor dimerization has an influence on the fate of the activated receptors.

Although the introduction of the manuscript is very well written, and makes a very compelling case for the story, my assessment is that the experimental data reported support these conclusions only to a limited extent, and more experimental data should be gathered to fully support the conclusions. This can be ascribed to methodological considerations, data analysis and pharmacological reasons that I will discuss here in detail.

The manuscript articulates around 6 main figures: the first figure introduces the methodological approach (first developed by Briane et al) based on the segmentation of individual trajectories into tracklets, which are in turn classified into the three categories of free diffusion, confined diffusion and directed motion. Figure 2 displays the outcome of this approach when applied to approx. 20000 tracks of wt CCR5, in basal conditions, showing that most tracklets (80%) display Brownian motion, and only 20% display restricted motion. We can define the outcome of the breakdown and categorization of tracklet motions as the dynamic fingerprint of the receptor. Figure 3 illustrates the effect of ligands on the dynamic fingerprint.

The authors use two agonists (CCL4 and PSC-RANTES) and one antagonist (maraviroc, MVC), to show that PSC-RANTES (10 minutes incubation) is able to significantly shift the fingerprint towards more immobile CCR5, whereas the antagonist increases the percentage of freely diffusing tracklets. Given the timescale after ligand addition, these results are compatible respectively with an increase and a decrease of receptor endocytosis, which is not unexpected.

In Figure 4 the authors explore the effect of Gi (by using PTX to uncouple) and β arrestin (Si-RNA) in modulating their dynamic fingerprint. While the effect of PTX is harder to dissect, knocking down β-arrestin expression clearly impacts the ability of the receptor to become 'restricted' in its motion, again consistent with the expected role of β-arrestin in initiating receptor endocytosis. Figure 4 brings about the role of dimerization in modulating CCR5 dynamic fingerprint. By switching to the analysis of the intensity of individual spots, rather than of their dynamics, the authors now compare the wt CCR5 to a dimerization impaired mutant, the L196K. The notable finding here is that the application of the antagonist increases the dimeric population in the dimerization impaired mutant, which in turn displays also a more restricted dynamic behavior. Such restricted dynamic behavior is also increased by PTX treatment.

Here, in summary, it appears that G protein uncoupling (either by PTX treatment and/or antagonist treatment) leads to a more restricted dynamic fingerprint of CCR5, which is a novel and somewhat counterintuitive result.

Finally, the effect of distinct HIV-1 capsid glycoproteins is studied in their agonistic capacity against CCR5 (Figure 6), with one gp (#34) having a marked effect increasing the number of restricted tracks in both wt and monomerized receptor mutant.

Overall, these data are used to support several conclusions by the authors, namely that (i) receptors switch mode of motion over time in response to changes in their molecular conformation, (ii) that this process is ligand dependent, (iii) HIV-1 gps act as (biased) agonists and (iv) dimerization ultimately regulates agonist response and subsequent endocytosis.

This review aims at verifying if the experimental data provided by the authors confirm the hypotheses and support the authors conclusion that dimerization of CCR5 plays an important role in modulating precoupling of the receptor to G protein, as well as the fate of individual receptors upon activation.

Methods: the authors make the significant choice of using EGFP N-terminally tagged receptors, instead of using cell lines with SNAP-tagged receptors (although available to the authors). The use of EGFP as opposed to an organic dye limits the photon budget significantly, and it is honestly unclear how trajectories with more than a few frames can be obtained. It is experience of the reviewer and several of his colleagues that tracks with more than 5-10 frames using fluorescent proteins are quite uncommon. At the (rather high) particle density displayed by the authors in their SI movies, there is the likelihood that the long trajectories observed stem from relinking of trajectories belonging to distinct receptors. Also, the notion that receptors conformational transitions reflect on receptor dynamics is extremely suggestive, but would definitely require a more stringent validation, for example combining SMT with a conformational sensor, i.e. by single molecule FRET. There are several inconsistencies in the way the data are presented. As a prominent example, the effect of PTX in the barchart of Figure 4A does not appear to be consistent with the time 0 behavior of panel 4B.

Moreover, the density of the molecules in the examples displayed raises questions on how a pure intensity analysis of the spot may accurately separate dimers from monomers. It would be paramount to rule out other effects that could cause the difference in intensity values and shapes of the histrograms in Figure 5 A, such as receptor expression levels. Also, other approaches to validate dimerization/monomerization such as molecular brightness could be easily implemented and should be used to validate the results.

Pharmacology: The dynamic behaviour of the receptor (% of restricted tracklets) in response to ligands is generally not surprising, and consistent with the receptor becoming immobilized in clathrin pits and being internalized over a timescale of several minutes.

The key novel and counterintuitive finding concerns the increase in restricted motion of the L196K mutant upon antagonist stimulation, although there is no hard evidence that this effect is strictly caused by the basal monomeric behavior of the receptor. Regarding effects on G protein coupling, please see my comment above about inconsistencies within PTX data.

Overall, the manuscript has a very ambitious and interesting scope, that would be better served, in my humble opinion, by several manuscripts(!), each dealing with individual aspects of this story, but going at a higher level of detail. The authors could use several methods to validate key findings (see comment above, for example, concerning dimerization).

I have the following key recommendation for the authors:

1. Repeat the experiments (for validation) using SNAP-tagged receptors, at lower density, and in a cell line with better membrane adhesion to the coverslip than HEK293. An option would be the HEK293AD variant, which typically displays a nice and extended basolateral membrane.

2. Check thoroughly the PTX data, since the data in Figure 4A and 4B appear inconsistent

3. Validate the dimerization data using a second, independent method

*Reviewer #2 (Recommendations for the authors):*

The manuscript by Momboisse et al., "Single molecule imaging reveals distinct effects on ligands on CCR5 dynamics depending on its dimerization status" seeks to understand the mobility, oligomerization and internalization parameters of the CCR5 receptor on the surface of live cells using total internal reflection microscopy. In this context, the manuscript specifically seeks to examine how these parameters are affected by chemokines and other extracellular interacting partners relevant to receptor regulation. These variables include investigations of pertussis toxin (PTX), PCS-RANTES and two solubilized forms of the gp120 heterotrimer derived from patients. The manuscript also describes analogous/parallel studies in the context of a previously characterized 'dimerization-compromised mutant (L196K)' as a means to examine whether CCR5 dimerization plays a critical role in the aforementioned parameters, downstream signal propagation, including receptor internalization.

The motivation for the proposed investigations is quite clear – although some of the language used to delineate the contributions/importance of conformational diversity to regulation – intermingled with compositional diversity – are vague in nature such that it is difficult for the reader to understand the underlying complexities in meaningful detail. The experimental variables examined are quite appropriate and appropriately considered with respect to exploring the relationship between CCR5 track dynamics (Brownian, Fixed or Mixed) and CCR5 dimerization and downstream signaling propensities. For the dimerization component of the investigations, the reader is left to believe solely on the basis of previously literature that the L196K mutant specifically/only effects the dimerization propensity of CCR5 and has no impact on any aspect of agonist-induced conformational response or downstream signaling. More clarity on what this mutation specifically effects would be helpful to the reader. While it seems a number of clarifications are required, the overarching finding is that eGFP-tagged CCR5 exhibits distinct types of dynamics ranging from highly mobile to highly fixed (no mobility), where in the fixed cellular domains CCR5 appears to cluster in a manner that the authors speculate relates to endocytosis. This overall take home message seems consistent with prior literature and knowledge in the field.. The finding that CCR5 exhibits distinct signaling propensities and interaction partners dependent on these dynamic attributions and its conformational states seems logical given previous findings, but less well-supported by the data presented.

For their live-cell tracking studies, the authors use a previously described eGFP-CCR5 construct that was a gift from F. Perez, originally reported in Boncompain et al. 2019. While eGFP has been used by multiple groups in the past to track the oligomerization status and mobility of cellular protein components (see for instance Iino, Moerner et al. BPJ 2001; Ulbrich and Isacoff NMETH 2007 and; Cui et al. Molecular Plant 2018), investigations into the fluorescent properties of eGFP that govern its utility for the types of tracking experiments performed (see for instance Peterman et a. J. Phys Chem A 1999 and Vamosi et al. Scientific Reports 2016) have generally led to the realization that confidence in quantitative tracking outputs from such experiments are prone to compromises that have motivated the field to move in the direction of increasing the signal-to-noise ratio of this type of study to quantum dot (see for instance Veya et al. JBC 2015) or organic fluorophore-labeled species (see for instance, Sako and Yanagita et al. Nat Cell Biol. 2000; Hern et al. PNAS 2010; Calebiro et al. PNAS 2013; Moller et al. Nat Chem Biol. 2020; Asher et al. NMETH 2021). From my review of the literature, it seems that eGFP is not always immediately matured (ie. fluorescent) in the cell and at the cell surface, it is prone to intermittent blinking and rapid photobleaching under conditions of illumination and acquisition frame rates employed in the present investigations. eGFP is also quite dim by comparison to organic fluorophores due principally to a relatively low absorbance cross section. For all these reasons, the total photon budget of eGFP is relatively low and variable, factors that can make robust tracking quite challenging. For instance, in the references cited above, it appears that eGFP has a lifetime of roughly 180 ms at 5 kW/cm2 (the precise illumination intensity used in the present investigations in not clear). At 100 ms integration time eGFP brightness per frame is about 12-fold above background. A commensurately lower total photon budget and signal above background is therefore expected at 33 ms integration time used in the present investigation. At higher illumination intensities, the results of Vamosi et al. (Sci. Report (2016)) would contend that further enhancements in eGFP brightness and photo budget are unlikely. Based on the data provided in the manuscript it is not possible to know what the quality of monomeric eGFP-CCR5 are: how bright the fluorophore is during the tracks; the variance in brightness during the tracks; how long the tracks are both in terms of time and total photon budget. This information should be reported as photons as opposed to AU. The incident illumination intensity should be reported in kW/cm2 so that others can reproduce the conditions, rather than a percent of the potential power output at the laser head. All this information seems essential for reader comprehension. Without such information, the aforementioned considerations from prior literature give pause to the believability behind the fundamental assumption that monomeric species are being tracked and that robust quantitative results have been obtained. Supporting the idea that eGFP may present challenges as a fluorophore, a recent paper by Li et al. "Oligomerization-enhanced receptor-ligand binding revealed by dual-color simultaneous tracking in living cell membranes", (J Phys Chem Lett, 2021) (which used eGFP in their prior investigations), seemingly relevant to the present investigations replaced eGFP on their CCR5 expression construct with mNeonGreen, stating than mNeonGreen is about 3-fold brighter than eGFP and exhibits a nearly 3-fold shorter maturation time. These concerns speak to the need for careful consideration and delineation of the details underlying the experiments, which at present appear to be absent form this version of the manuscript. Consequently, my interpretation of the results started with examination of the construct examined to determine if it was an oligomeric eGFP construction being used.

After digging a bit through the Boncompain manuscript (the original paper describing the eGFP-CCR5 construct), I was not able to discern the precise nature of the eGFP-tagged CCR5 construct that was utilized in the present investigation. This consideration speaks to the overall paucity of information provided by the authors as to the underlying details of the experiments performed that would enable reproducibility. As far I can discern, the construct employed is based on a "RUSH" plasmid in which a streptavidin binding protein (SBP) is fused first to eGFP and then to CCR5 (N- to C-terminus) and co-expressed with an ER-resident protein (KDEL)-fused to streptavidin. In this assay, the plasmid containing this dual expressing system "Str-KDEL_SBP-EGFP-CCR5" is first transfected to obtain a stable cell line and then the stable cell line constitutively expresses the SBP-eGFP-CCR5 construction such that it is sequestered in the endoplasmic reticulum (ER) until the addition of biotin. Biotin releases the streptavidin-sequestered eGFP-CCR5 molecules trapped in the ER by outcompeting the SBP interaction. Biotin releases the SBP-eGFP-CCR5 protein to the cell surface through a Golgi apparatus-mediated secretion pathway. I was unable to find any mention of biotin in the manuscript so additional details about the stable cell line construction and protein expression seem relevant to include.

Given the data presented, it is not entirely clear what state of oligomerization the expressed eGFP-tagged CCR5 protein is in upon reaching the cell surface, or the tracks that being quantified. The authors state that CCR5 is naturally found in a variety of homo- and hetero-oligomeric complexes and hence, it is not at all clear what the precise composition is of the "particles" – homomeric, heteromeric or otherwise – being tracked. Are these considerations not important to the interpretations of their findings? It certainly seems relevant to take these considerations into account in regard to the conveyance of quantitative interpretations to the reader of the underlying biology.

In the vein of increased precision, the authors may also wish to temper their claims about the "original" or "novel" nature of their 'quantitative' tracking approach. If I am understanding things correctly, the acquisition of the tracks is based on the software provided in https://www.cell.com/cell-reports/pdf/S2211-1247(18)30071-8.pdf, where the classification of the tracklets (individual tracks computationally parsed into 5 frames) relies on a statistical tool initially developed in https://hal.inria.fr/hal-01416855/document and later published in https://arxiv.org/pdf/1804.04977.pdf. On this point, it is somewhat unclear to me that the 'innovation' of segmenting tracks into 5 frames using a software that was constructed and optimized using tracks of 30 frames (if I understand things correctly), is without hazard. Commentary on why the chosen analysis method(s) may be superior to other analysis approaches also seems worthy of comment.

In this light, it seems excessive to repeatedly mention the originality of their SPT tool. Moreover, as for the shortage of references to prior literature that have attempted to perform live-cell tracking of single-molecules in the membrane, the authors should be aware that the processing approaches employed have been implemented for many years by multiple groups. See for instance:

https://ani.stat.fsu.edu/~hycao/docs/pub/biometrika_15.pdf

https://www.nature.com/articles/nmeth.3483

https://journals.plos.org/plosone/article?id=10.1371/journal.pone.0082799

https://hal.inria.fr/hal-01416855/document

https://arxiv.org/pdf/1804.04977.pdf. (cited in the paper).

https://www.cell.com/biophysj/pdfExtended/S0006-3495(18)30132-2

https://doi.org/10.1016/j.ymeth.2020.03.008

https://iopscience.iop.org/article/10.1088/1478-3975/ab64b3

https://www.nature.com/articles/nmeth.2808

The specific novelty of the present investigations relates to the choice of parameters (N and α) that are required for the classification of the tracklets (pieces of tracks acquired via ICY software as it is introduced in https://arxiv.org/pdf/1804.04977.pdf). In the paper, the authors made the choice of N=5 and the choice of α = 0.05. The authors justify the use of these chosen parameter values in the supplementary material using statistical tests to examine the robustness/validity of these choices. However, the simulated tracks generated for this test, which exhibit the specific motion models that they aim to differentiate in live cells (Brownian, fixed or mixed), do not seem to be similar to the tracks actually examined in the authors' experiments. Specifically, the authors do not appear to have incorporated any measurement noise in their tracks, including background noise fluctuations, eGFP signal variances, blinking etc. At the particle densities shown in the SI movies, it also seems unavoidable that errors will be made when particle tracks cross paths. This validity test therefore seems to be quite unrealistic as it pertains to confidently discerning quantitative aspects of the underlying biology. For the reader to have confidence in the results presented, it would seem imperative to provide actual eGFP tracks to the reader in terms of photons/AU per frame as a function of time and provide some estimate of how relevant the chosen parameters are from their simulations to the actual experimental setting where the signal-to-noise ratio is much lower presumably and the tracks are potentially more crowded within the cell-surface area. What errors in the authors' measurements may be associated with such considerations?

Given these considerations, while the overall goals of the work are important and the results appear globally consistent with prior literature, without clarifications to issues delineated above, it is difficult to ascertain the validity of the authors' overall interpretation that CCR5 can dimerize in a manner that directly relates to G-protein/arrestin signaling and to what quantitative extent dimerization contributes to these events or the subsequent process of internalization. In the absence of significant efforts to address the concerns raised, the authors may wish to temper their claims and conclusions substantially and address their observations and qualitatively supportive of the models proposed, which do seem reasonable in nature and globally consistent with expectation.

*Reviewer #3 (Recommendations for the authors):*

Investigating the dynamic interaction of GPCRs on the surface of cells is of great importance for improving our understanding of signaling across the plasma membrane. Momboisse et al. use TIRF based single-molecule imaging to track the movement of individual CCR5 molecules under different conditions. They use a different tracking statistics method that enables the analysis of motion changes. They show that treating cells with agonists results in more restricted movements and clustering of receptors as well as receptor endocytosis. Responsible for these effects is the interaction with β-arrestin. Inverse agonist have the opposite effect while binding of the HIV-1 envelope glycoprotein gp120 shows agonist-like properties.

Important for the immobilization of the receptor is its dimerization (and to a small percentage formation of higher oligomers).

The reported results are a nice example of the quantitative analysis of the movement of cell surface receptors and will be of interest for the analysis of other GPCRs and cell surface receptors as well.

1) Heterodimerization with other GPCRs (e.g. CCR2) are documented. Can it be excluded that for example the percentage of receptors showing restricted movement in non stimulated cells is due to such oligomerizations? CCR2 or other receptors would not be fluorescence labeled and could therefore not be detected.

2) The clustering of arrestin has been documented and discussed in the two following manuscripts:

1) Coordinate-based co-localization-mediated analysis of arrestin clustering upon stimulation of the C-C chemokine receptor 5 with RANTES/CCL5 analogues.

Tarancón Díez L, Bönsch C, Malkusch S, Truan Z, Munteanu M, Heilemann M, Hartley O, Endesfelder U, Fürstenberg A. Histochem Cell Biol. 2014 Jul;142(1):69-77. doi: 10.1007/s00418-014-1206-1. Epub 2014 Mar 13. PMID: 24623038

2) Quantitative morphological analysis of arrestin2 clustering upon G protein-coupled receptor stimulation by super-resolution microscopy.

Truan Z, Tarancón Díez L, Bönsch C, Malkusch S, Endesfelder U, Munteanu M, Hartley O, Heilemann M, Fürstenberg A. J Struct Biol. 2013 Nov;184(2):329-34. doi: 10.1016/j.jsb.2013.09.019. Epub 2013 Sep 30. PMID: 24091038

These should be cited and the results being included in the discussion.

---

## [Author Response]

[Editors’ note: The authors appealed the original decision. What follows is the authors’ response to the first round of review.]

Reviewer #1 (Recommendations for the authors):The manuscript from Momboisse et al. provides a single molecule tracking (SMT) investigation of the CCR5 receptor on the surface of living cells.[…]Overall, these data are used to support several conclusions by the authors, namely that (i) receptors switch mode of motion over time in response to changes in their molecular conformation, (ii) that this process is ligand dependent, (iii) HIV-1 gps act as (biased) agonists and (iv) dimerization ultimately regulates agonist response and subsequent endocytosis.This review aims at verifying if the experimental data provided by the authors confirm the hypotheses and support the authors conclusion that dimerization of CCR5 plays an important role in modulating precoupling of the receptor to G protein, as well as the fate of individual receptors upon activation.Methods: the authors make the significant choice of using EGFP N-terminally tagged receptors, instead of using cell lines with SNAP-tagged receptors (although available to the authors). The use of EGFP as opposed to an organic dye limits the photon budget significantly, and it is honestly unclear how trajectories with more than a few frames can be obtained. It is experience of the reviewer and several of his colleagues that tracks with more than 5-10 frames using fluorescent proteins are quite uncommon. At the (rather high) particle density displayed by the authors in their SI movies, there is the likelihood that the long trajectories observed stem from relinking of trajectories belonging to distinct receptors. Also, the notion that receptors conformational transitions reflect on receptor dynamics is extremely suggestive, but would definitely require a more stringent validation, for example combining SMT with a conformational sensor, i.e. by single molecule FRET.

Observed long trajectories with eGFP tagged receptors do not necessarily correspond to particle relinking as we only considered tracks with relevant signal-to-noise ratio (SNR) and few missed detections and used a very moderate illumination intensity (0.4 kW/cm^2^) over short periods (3-6 seconds) to decrease bleaching.

That said, and at high particles’ density, there is indeed a risk of mistracking and trajectories’ relinking. To mitigate that risk, we fragmented receptors’ tracks into very short tracklets (N=5 frames). To assess the robustness of our classification method with respect to the density of tracked particles, we performed a novel set of simulations where we simulated *n=1000* confined tracklets (N=5 frames) and generated the corresponding synthetic movies with a mixed Poisson-Gaussian noise model (see Material and Methods) for 3 different spots’ densities (d=0.039, 0.16 and 0.63 spots/μm2, the measured density being <0.5 spots/μm2 in most experiments). For different percentages of confined tracklets (10 % confined & 90% Brownian; 50 % confined & 50% Brownian and 100 % confined) we observed that the classification accuracy was robust to increased densities of spots (Author response image 1), which assessed the robustness of our algorithm in experimental conditions (Figure 1, figure supplement 1) (page 7).

**Author response image 1. sa2fig1:** Relative accuracy of statistical classification for increasing spots’ density. The proportion of confined trajectories was set to 10% (90% Brownian, blue), 50% (50% Brownian, red) and 100% (0% Brownian, green). Confinement parameter was fixed to *λ* = 2. For each condition (density and proportion of confined trajectories), we run n = 10 simulations with 1000 moving spots in a 1600x1600 pixels (density = 0.039 spots/μm^2^), 800 X 800 pixels (density = 0.16 spots/μm^2^) and 400 X 400 pixels (density = 0.63 spots/μm^2^).

In addition, to confirm our findings and to avoid a bias related to a unique model, we have added a new dataset carried out in another model, in which we tracked FLAG-SNAP-tagged proteins using fluorescent antibodies. The results obtained with this second model (FLAG-SNAP-tagged-CCR5) confirmed those obtained with the first one (eGFP-CCR5) in the different conditions tested (see details below in the “recommendation to authors” section).

There are several inconsistencies in the way the data are presented. As a prominent example, the effect of PTX in the barchart of Figure 4A does not appear to be consistent with the time 0 behavior of panel 4B.

We corrected Figure 4B (in presence of PSC-RANTES) by adding the time 0, which was missing. As for Figure 4A (basal state), we observed at t0 a small but significant decrease in CCR5 mobility restriction in the presence of PTX compared to untreated cells. This suggests that a small fraction of CCR5 is in Gi protein-bound form in the basal state

Moreover, the density of the molecules in the examples displayed raises questions on how a pure intensity analysis of the spot may accurately separate dimers from monomers. It would be paramount to rule out other effects that could cause the difference in intensity values and shapes of the histrograms in Figure 5 A, such as receptor expression levels. Also, other approaches to validate dimerization/monomerization such as molecular brightness could be easily implemented and should be used to validate the results.

We agree that a fluorescence intensity analysis of the spot may not be sufficient to accurately discriminate monomers from dimers. We attributed the fluorescence intensities of the 3 types of Gaussians observed in eGFP-CCR5-WT expressing cells to monomers/dimers/oligomers based on the fluorescence intensity of eGFP spotted on coverslip, which quenched in a single step (Salavessa, 2021).

However, we are aware that this fluorescence intensity measurement is relative to eGFP on coverslip. Therefore, we are now describing the different receptor forms as « low », « medium » or « high » fluorescence intensity forms instead of monomers/dimers/oligomers (page 17). As a consequence, we changed the title, by removing the term “single molecule tracking”.

However, regardless of the exact amounts of individual molecules in the “low”, “medium” or “high” fluorescence populations, their relative proportions differed between CCR5-WT and the dimerization-compromised mutant CCR5-L196K. We previously showed by alternative approaches (molecular modeling, HTRF, and a functional assay) (Jin, 2018) that this mutant formed less dimers and higher order oligomers than CCR5-WT. This is in agreement with the results in Figure 5A showing a reduction in the proportions of the “high” and “medium” fluorescence populations for eGFP-CCR5-L196K in favor of the “low” fluorescence population, relative to CCR5-WT. Figure 5B also showed that the CCR5 inverse agonist MVC increased the proportion of CCR5-L196K populations with “high or medium” fluorescence intensity. Again, this is in accordance with our previous work showing that MVC can stabilize CCR5-L196K in a dimeric conformation (Jin, 2018). Considered altogether, these data indicate that our classification method based on fluorescence intensity analysis of receptor populations is accurate to characterize the heterogeneity of receptor organization at the cell surface.

Characterizing the « exact » stoichiometry of molecules per spot is another question, highly challenging, which we will examine in a future study using molecular brightness and super-resolution approaches.

Of note, to minimize the influence of the amount of surface receptor on the stoichiometry of the molecule/spot, we worked with clones expressing similar amounts of cell surface receptors as mentioned page 17, line 460.

Pharmacology: The dynamic behaviour of the receptor (% of restricted tracklets) in response to ligands is generally not surprising, and consistent with the receptor becoming immobilized in clathrin pits and being internalized over a timescale of several minutes.The key novel and counterintuitive finding concerns the increase in restricted motion of the L196K mutant upon antagonist stimulation, although there is no hard evidence that this effect is strictly caused by the basal monomeric behavior of the receptor. Regarding effects on G protein coupling, please see my comment above about inconsistencies within PTX data.

As mentioned by the reviewer, the dynamic behavior of eGFP-CCR5-WT upon activation supports receptor trapping in nanodomains. However, contrary to what is proposed by the reviewer, receptor clustering and massive immobilization is not an absolute prerequisite for receptor endocytosis. Indeed, we showed that CCR5-L196K, although impaired in agonist-dependent immobilization (Figure 5E, F), successfully recruits b-arrestins in response to agonist binding and is internalized (see below and new Figure 6). This suggests that receptor immobilization, triggered by barrestin recruitment, requires receptor oligomerization.

On one hand, our data showed a change in CCR5-WT mobility upon agonist activation towards immobilization, while treatment with an inverse agonist had the opposite effect (Figure 3). PTX treatment and b-arresting silencing suggested that this effect was not Gi-protein-dependent but dependent on b-arrestins recruitment (Figure 4).

On the other hand, results with the dimerization-compromised mutant CCR5-L196K showed that CCR5 clustering and immobilization was not a pre-requisite to its internalization. Indeed, contrary to eGFP-CCR5-WT massive immobilization, eGFP-CCR5-L196K was only slightly immobilized after 10 min of agonist treatment (Figure 5 E-F). In this condition, the internalization process was less efficient but not abrogated (Figure 6C), which is counterintuitive. We assumed that CCR5-L196K may lead to a conformation that recruits little or no b-arrestins or that interacts differently with b-arrestins. To test this hypothesis, we have now added a new set of experiments to evaluate agonist-induced barr2 recruitment. We showed that b-arrestins is correctly recruited to the plasma membrane upon CCR5-L196K activation (new Figure 6A, B). This suggested that the lack of immobilization and clustering of CCR5-L196K is not due to a default of b-arrestins recruitment. We propose that the conformational organization of the receptor and in particular its oligomeric status is necessary for the b-arrestin to trigger receptor clustering and optimize receptor internalization.

These findings are presented page 20 and discussed page 28.

We agree that multiple factors may come into play to explain the lack of immobilization of activated CCR5-L196K. However, we showed previously that CCL3 binds CCR5-L196K and CCR5-WT with similar affinities (Jin, 2018; Colin, 2018). In addition, we have added new results showing that activated CCR5-WT and CCR5-L196K similarly transduced ERK signaling (Figure 5—figure supplement 1). That is why we attributed the lack of CCR5-L196K immobilization after stimulation to its altered capacity to forms dimers, and not to an effect on ligand binding or signaling.

I have the following key recommendation for the authors:1. Repeat the experiments (for validation) using SNAP-tagged receptors, at lower density, and in a cell line with better membrane adhesion to the coverslip than HEK293. An option would be the HEK293AD variant, which typically displays a nice and extended basolateral membrane.

We carried out another set of experiments using a second model in which we tracked FLAG-SNAP-tagged proteins using fluorescent antibodies. The results obtained with this second model (FLAG-SNAP-tagged-CCR5) confirmed those obtained with the first one (eGFP-CCR5) in the different conditions tested: after ligand binding (using two agonists, an inverse agonist, and HIV-1 envelope glycoproteins) and after modulation of receptor dimerization (using the eGFP-CCR5-L196K mutant). They validated our tracking analysis and our conclusions, supporting that our findings are independent of the model used. We added 4 Figures as Figure supplements and 1 video (Video 4, Figure 2—figure supplement 1, Figure 3, figure supplement 2, Figure 5—figure supplement 2, Figure 7—figure supplement 1). We mentioned the results in the manuscript pages 6, 10, 13, 18, 22.

2. Check thoroughly the PTX data, since the data in Figure 4A and 4B appear inconsistent.

As mentioned above, we corrected Figure 4B by adding the time 0, which was missing.

3. Validate the dimerization data using a second, independent method.

As mentioned above, we agree that the fluorescence intensity of the spots cannot accurately validate the stoichiometry of the molecules per spot. We are now describing the different receptor forms by their relative fluorescent intensity (« low », « medium » or « high » fluorescent intensity forms) compared to eGFP spotted on coverslip instead of monomers/dimers/oligomers (page 17).

This approach, even relative, confirmed that CCR5-L196K forms fewer high order oligomers than CCR5-WT. This is fully consistent with our previous work using alternative approaches (molecular modeling, FRET, and a functional assay), and showing that the L196K mutation in the CCR5 dimerization interface impairs the receptor in its capacity to form di-/oligo-mers (Jin, 2018).

Reviewer #2 (Recommendations for the authors):[…]The experimental variables examined are quite appropriate and appropriately considered with respect to exploring the relationship between CCR5 track dynamics (Brownian, Fixed or Mixed) and CCR5 dimerization and downstream signaling propensities. For the dimerization component of the investigations, the reader is left to believe solely on the basis of previously literature that the L196K mutant specifically/only effects the dimerization propensity of CCR5 and has no impact on any aspect of agonist-induced conformational response or downstream signaling. More clarity on what this mutation specifically effects would be helpful to the reader.

We are now describing in more details the influence of the mutation L196K on the functional properties of CCR5 (page 17).

Previously, we showed that Leu-196 in TM5 is part of the CCR5 dimerization interface. Its substitution by Lys in CCR5-L196K inhibits the capacity of the receptor to dimerize, as revealed by complementary approaches such as molecular modeling, energy transfer, and a functional export assay (Jin, 2018).

Functionally, the L196K mutation decreases cell surface expression of the receptor due to retention in the endoplasmic reticulum (Jin, 2018). However, CCR5-L196K folding is not altered as shown by its ability to bind chemokines and HIV-1 gp120s with the same affinity as CCR5-WT (Jin, 2018; Colin, 2018). Additionally, in cells expressing similar amounts of CCR5-WT or CCR5-L196K, we now showed that both receptors similarly activate ERK1/2 after agonist stimulation (Figure 5—figure supplement 1).

Finally, our manuscript indicated (i) that receptor immobilization depended on b-arrestin recruitment (Figure 4) and (ii) that only receptors able to oligomerize were immobilized upon activation (Figure 5). We now added a set of data revealing that CCR5-L196K and CCR5-WT similarly recruited b-arrestins (Figure 6A, B) and was internalized upon activation (Figure 6C). This suggested that b-arrestin-induced receptor immobilization requires receptor oligomerization and that receptor immobilization is not a prerequisite for receptor endocytosis. However, CCR5-L196K internalization was less efficient compared to CCR5-WT, suggesting that receptor immobilization and clustering are important for an optimal endocytosis.

While it seems a number of clarifications are required, the overarching finding is that eGFP-tagged CCR5 exhibits distinct types of dynamics ranging from highly mobile to highly fixed (no mobility), where in the fixed cellular domains CCR5 appears to cluster in a manner that the authors speculate relates to endocytosis. This overall take home message seems consistent with prior literature and knowledge in the field.

As explained above, our experiments suggest that massive immobilization may not be a prerequisite for endocytosis. Indeed, CCR5-L196K remains highly mobile after agonist stimulation, while being able to be internalized and to recruit b-arrestins. As such, we believe that our results provide an additional degree of information compared to the current literature.

The finding that CCR5 exhibits distinct signaling propensities and interaction partners dependent on these dynamic attributions and its conformational states seems logical given previous findings, but less well-supported by the data presented.

In the present manuscript, we showed that CCR5-WT mobility at the plasma membrane upon PSC-RANTES activation depends on b-arrestins (and not on Gi-protein) (Figure 4), a feature only recently reported after MOR stimulation (Markova, 2021).

The use of a dimerization-compromised mutant added that the mobility of the receptor and its optimal internalization depend on receptor dimerization. In particular, we propose that b-arrestin-induced receptor immobilization requires receptors oligomerization.

For their live-cell tracking studies, the authors use a previously described eGFP-CCR5 construct that was a gift from F. Perez, originally reported in Boncompain et al. 2019. While eGFP has been used by multiple groups in the past to track the oligomerization status and mobility of cellular protein components (see for instance Iino, Moerner et al. BPJ 2001; Ulbrich and Isacoff NMETH 2007 and; Cui et al. Molecular Plant 2018), investigations into the fluorescent properties of eGFP that govern its utility for the types of tracking experiments performed (see for instance Peterman et a. J. Phys Chem A 1999 and Vamosi et al. Scientific Reports 2016) have generally led to the realization that confidence in quantitative tracking outputs from such experiments are prone to compromises that have motivated the field to move in the direction of increasing the signal-to-noise ratio of this type of study to quantum dot (see for instance Veya et al. JBC 2015) or organic fluorophore-labeled species (see for instance, Sako and Yanagita et al. Nat Cell Biol. 2000; Hern et al. PNAS 2010; Calebiro et al. PNAS 2013; Moller et al. Nat Chem Biol. 2020; Asher et al. NMETH 2021). From my review of the literature, it seems that eGFP is not always immediately matured (ie. fluorescent) in the cell and at the cell surface, it is prone to intermittent blinking and rapid photobleaching under conditions of illumination and acquisition frame rates employed in the present investigations. eGFP is also quite dim by comparison to organic fluorophores due principally to a relatively low absorbance cross section. For all these reasons, the total photon budget of eGFP is relatively low and variable, factors that can make robust tracking quite challenging. For instance, in the references cited above, it appears that eGFP has a lifetime of roughly 180 ms at 5 kW/cm2 (the precise illumination intensity used in the present investigations in not clear). At 100 ms integration time eGFP brightness per frame is about 12-fold above background. A commensurately lower total photon budget and signal above background is therefore expected at 33 ms integration time used in the present investigation. At higher illumination intensities, the results of Vamosi et al. (Sci. Report (2016)) would contend that further enhancements in eGFP brightness and photo budget are unlikely. Based on the data provided in the manuscript it is not possible to know what the quality of monomeric eGFP-CCR5 are: how bright the fluorophore is during the tracks; the variance in brightness during the tracks; how long the tracks are both in terms of time and total photon budget. This information should be reported as photons as opposed to AU. The incident illumination intensity should be reported in kW/cm2 so that others can reproduce the conditions, rather than a percent of the potential power output at the laser head. All this information seems essential for reader comprehension. Without such information, the aforementioned considerations from prior literature give pause to the believability behind the fundamental assumption that monomeric species are being tracked and that robust quantitative results have been obtained. Supporting the idea that eGFP may present challenges as a fluorophore, a recent paper by Li et al. "Oligomerization-enhanced receptor-ligand binding revealed by dual-color simultaneous tracking in living cell membranes", (J Phys Chem Lett, 2021) (which used eGFP in their prior investigations), seemingly relevant to the present investigations replaced eGFP on their CCR5 expression construct with mNeonGreen, stating than mNeonGreen is about 3-fold brighter than eGFP and exhibits a nearly 3-fold shorter maturation time. These concerns speak to the need for careful consideration and delineation of the details underlying the experiments, which at present appear to be absent form this version of the manuscript. Consequently, my interpretation of the results started with examination of the construct examined to determine if it was an oligomeric eGFP construction being used.

We thank the reviewer for all of these constructive comments.

We have now specified the illumination intensities (< 0.4 kW/cm^2^ for tracking; < 0.7 kW/cm^2^ for fluorescence intensity) (page 37). We measured the power at the output of the microscope related to the smallest illumination disk corresponding to the field of the camera. Note that the sample being larger than the field of the camera, the measured values were probably overestimated. The intensity values measured were 10 times lower than those mentioned in the article cited by the reviewer. The camera detector used in our experiments is likely much more sensitive than that used several years ago in the cited article.

Under this relatively low illumination power, the intensity of the spots is stable throughout the duration of the acquisition (200 frames) (Author response image 2 “mean intensities / frames”) allowing the tracking of long trajectories. In addition, we only considered tracks with relevant SNR and few missed detections. To mitigate a risk of mistracking and trajectories relinking, we also fragmented receptors’ tracks into very short tracklets (N=5 frames) (Materials et Methods pages 40-41).

As mentioned above in response to the reviewer 1, we considered that the fluorescence intensity analysis of the spot is relative to eGFP spotted on coverslip and is not sufficient to accurately discriminate monomers from dimers. Therefore, we are now describing the different receptor forms as « low », « medium » or « high » fluorescence intensity forms instead of monomers/dimers/oligomers (pages 17). Since we are not quantifying single particles, the photons/AU parameter is not relevant here.Finally, as mentioned in response to the reviewer 1, we carried out another set of experiments using a second model in which we tracked FLAG-SNAP-tagged proteins using fluorescent antibodies. The results obtained with this second model (FLAG-SNAP-tagged-CCR5) confirmed those obtained with the first one (eGFP-CCR5) in the different conditions tested. They validated our tracking analysis and our conclusions, supporting that our findings are independent of the model used (Video 4, Figure 2—figure supplement 1, Figure 3, figure supplement 2, Figure 5—figure supplement 2, Figure 7—figure supplement 1).

After digging a bit through the Boncompain manuscript (the original paper describing the eGFP-CCR5 construct), I was not able to discern the precise nature of the eGFP-tagged CCR5 construct that was utilized in the present investigation. This consideration speaks to the overall paucity of information provided by the authors as to the underlying details of the experiments performed that would enable reproducibility. As far I can discern, the construct employed is based on a "RUSH" plasmid in which a streptavidin binding protein (SBP) is fused first to eGFP and then to CCR5 (N- to C-terminus) and co-expressed with an ER-resident protein (KDEL)-fused to streptavidin. In this assay, the plasmid containing this dual expressing system "Str-KDEL_SBP-EGFP-CCR5" is first transfected to obtain a stable cell line and then the stable cell line constitutively expresses the SBP-eGFP-CCR5 construction such that it is sequestered in the endoplasmic reticulum (ER) until the addition of biotin. Biotin releases the streptavidin-sequestered eGFP-CCR5 molecules trapped in the ER by outcompeting the SBP interaction. Biotin releases the SBP-eGFP-CCR5 protein to the cell surface through a Golgi apparatus-mediated secretion pathway. I was unable to find any mention of biotin in the manuscript so additional details about the stable cell line construction and protein expression seem relevant to include.

We did not use the SBP-eGFP-CCR5 plasmid used in Boncompain, 2019, but a different plasmid expressing eGFP-CCR5 from the CMV promoter. We have now clarified this point in the material and method section and have deleted the reference boncompain, 2019, so as not to mislead the reader (page 34).

Given the data presented, it is not entirely clear what state of oligomerization the expressed eGFP-tagged CCR5 protein is in upon reaching the cell surface, or the tracks that being quantified. The authors state that CCR5 is naturally found in a variety of homo- and hetero-oligomeric complexes and hence, it is not at all clear what the precise composition is of the "particles" – homomeric, heteromeric or otherwise – being tracked. Are these considerations not important to the interpretations of their findings? It certainly seems relevant to take these considerations into account in regard to the conveyance of quantitative interpretations to the reader of the underlying biology.

On one hand, we tracked receptor populations over time, and on the other hand, we analyzed the intensity of spots from frame 1. However, we did not make a link between fluorescence intensity and tracking over time. We therefore do not know how the populations evolve over time. This would require associating AIC algorithm with the tracking one. We will implement such a link in a near future. We consider here that studying the dimerization-compromised mutant, which we well characterized in our previous work (Jin, 2018), is sufficient to support a role of receptor dimerization in its dynamics behavior.

Fluorescence intensity analysis revealed different CCR5 homo-oligomeric organization at the cell surface. Studying CCR5 heterodimerization, which require to follow receptors expressing two different fluorescent proteins, goes beyond the scope of this study. Performing such a study will help understand the impact of the heterodimerization on receptor functions, which is of particular interest.

In the vein of increased precision, the authors may also wish to temper their claims about the "original" or "novel" nature of their 'quantitative' tracking approach. If I am understanding things correctly, the acquisition of the tracks is based on the software provided in https://www.cell.com/cell-reports/pdf/S2211-1247(18)30071-8.pdf, where the classification of the tracklets (individual tracks computationally parsed into 5 frames) relies on a statistical tool initially developed in https://hal.inria.fr/hal-01416855/document and later published in https://arxiv.org/pdf/1804.04977.pdf. On this point, it is somewhat unclear to me that the 'innovation' of segmenting tracks into 5 frames using a software that was constructed and optimized using tracks of 30 frames (if I understand things correctly), is without hazard. Commentary on why the chosen analysis method(s) may be superior to other analysis approaches also seems worthy of comment.In this light, it seems excessive to repeatedly mention the originality of their SPT tool. Moreover, as for the shortage of references to prior literature that have attempted to perform live-cell tracking of single-molecules in the membrane, the authors should be aware that the processing approaches employed have been implemented for many years by multiple groups. See for instance:https://ani.stat.fsu.edu/~hycao/docs/pub/biometrika_15.pdfhttps://www.nature.com/articles/nmeth.3483https://journals.plos.org/plosone/article?id=10.1371/journal.pone.0082799https://hal.inria.fr/hal-01416855/documenthttps://arxiv.org/pdf/1804.04977.pdf. (cited in the paper).https://www.cell.com/biophysj/pdfExtended/S0006-3495(18)30132-2https://doi.org/10.1016/j.ymeth.2020.03.008https://iopscience.iop.org/article/10.1088/1478-3975/ab64b3https://www.nature.com/articles/nmeth.2808The specific novelty of the present investigations relates to the choice of parameters (N and α) that are required for the classification of the tracklets (pieces of tracks acquired via ICY software as it is introduced in https://arxiv.org/pdf/1804.04977.pdf). In the paper, the authors made the choice of N=5 and the choice of α = 0.05. The authors justify the use of these chosen parameter values in the supplementary material using statistical tests to examine the robustness/validity of these choices. However, the simulated tracks generated for this test, which exhibit the specific motion models that they aim to differentiate in live cells (Brownian, fixed or mixed), do not seem to be similar to the tracks actually examined in the authors' experiments. Specifically, the authors do not appear to have incorporated any measurement noise in their tracks, including background noise fluctuations, eGFP signal variances, blinking etc. At the particle densities shown in the SI movies, it also seems unavoidable that errors will be made when particle tracks cross paths. This validity test therefore seems to be quite unrealistic as it pertains to confidently discerning quantitative aspects of the underlying biology. For the reader to have confidence in the results presented, it would seem imperative to provide actual eGFP tracks to the reader in terms of photons/AU per frame as a function of time and provide some estimate of how relevant the chosen parameters are from their simulations to the actual experimental setting where the signal-to-noise ratio is much lower presumably and the tracks are potentially more crowded within the cell-surface area. What errors in the authors' measurements may be associated with such considerations?

We thank the referee for having pointed out several relevant papers to our study. Many of them are actually based on the probabilistic framework of Bayesian analysis to determine the most likely repartition of trajectories into classes depending on their diffusion coefficient. While Bayesian analysis can be used to classify trajectories, we chose to rather implement a method based on statistical hypothesis testing as it is more efficient for short tracklets, does not requires an important computational load and can be easily implemented. We have now better discussed the choice of our classification method and added references on page 7.

The “novelty” of our approach relies on the choice of a classification pipeline, which consists here of splitting receptors’ tracks into very short tracklets (N=5 frames) to mitigate the risk of mis-tracking and dynamics’ change along each receptor’s trajectory, and implementing a robust statistical classifier initially developed by Kervrann and co-authors. We have therefore soften our claims throughout the manuscript by removing all the inappropriate “novel”… terms. In addition, and to motivate our approach, we have now performed new sets of simulations to validate our choice of very short tracklets (N=5), and assess the robustness of our statistical classifier with respect to image SNR and spots density. (Figure 1—figure supplement 1)

First, the accuracy of tracklet classification is expected to decrease with tracklet length. By using Monte-Carlo simulations of confined motion for an increasing confinement parameter λ (see Material and Methods), we compared the accuracy obtained for N=5 and N=10 tracklets. We observed that, indeed, the classification accuracy for N=5 is almost 2-fold decreased compared to N=10. Then, to account for the risk of mistracking, or a change in receptor dynamics, we modeled a generic *perturbation* in receptor tracking with a standard exponential distribution with rate ρ. Therefore, the conditional probability *p_N_(ρ)* for a confined tracklet with length N to be correctly classified is given by pN(ρ)=pN(0)exp⁡(−ρN), where pN(0) is the classification accuracy when no risk of mistracking or dynamic change is considered. We observed that for ρ=0.1, i.e. for a mean time of correct tracking/absence of dynamic change of 10 frames, the conditional probability of accurate classification for N=5 or N=10 converged to very similar values, and that for higher rate such as ρ=0.2, the conditional accuracy became greater for very short tracklets (N=5). In other words, when the rate of mistracking or dynamics change is quite high, which is likely the case in our experimental dataseet (low SNR and high density of particles), it is more accurate to consider very short tracklets.

Second, to assess the robustness of our classification method with respect to the density of tracked particles, we performed a novel set of simulations where we simulated *n=1000* confined tracklets (N=5 frames) and generated the corresponding synthetic movies with a mixed Poisson-Gaussian noise model (see Material and Methods) for 3 different spots’ densities (d=0.039, 0.16 and 0.63 spots/μm2, the measured density being <0.5 spots/μm2 in most experiments). For different percentage of confined tracklets (10 % confined / 90% Brownian ; 50 % confined / 50% Brownian and 100 % confined) we observed that the classification accuracy was robust to increased densities of spots, which validated the robustness of our algorithm.

Third, the noise (background noise, fluctuating intensity of particles…) within experimental movies could, indeed, affects the accuracy of tracklets classification because of the potential mis-localization of receptors’ spots (the localization accuracy depends on the SNR as discussed in Ober, R. J., Ram, S., & Ward, E. S. (2004). Localization accuracy in single-molecule microscopy. *Biophysical journal*, *86*(2), 1185-1200.). To assess the robustness of our classification method to noise, we performed a third set of simulations where we simulated confined tracklets and generated the corresponding synthetic movies for decreasing SNR. We observed that our method remained very accurate for SNR≥6, before decreasing due to spots’ mis-detection and localization. In our experimental dataset, we measured a mean SNR∼10, validating the robustness of our statistical classifier.

Reviewer #3 (Recommendations for the authors):Investigating the dynamic interaction of GPCRs on the surface of cells is of great importance for improving our understanding of signaling across the plasma membrane. Momboisse et al. use TIRF based single-molecule imaging to track the movement of individual CCR5 molecules under different conditions. They use a different tracking statistics method that enables the analysis of motion changes. They show that treating cells with agonists results in more restricted movements and clustering of receptors as well as receptor endocytosis. Responsible for these effects is the interaction with β-arrestin. Inverse agonist have the opposite effect while binding of the HIV-1 envelope glycoprotein gp120 shows agonist-like properties.Important for the immobilization of the receptor is its dimerization (and to a small percentage formation of higher oligomers).The reported results are a nice example of the quantitative analysis of the movement of cell surface receptors and will be of interest for the analysis of other GPCRs and cell surface receptors as well.1) Heterodimerization with other GPCRs (e.g. CCR2) are documented. Can it be excluded that for example the percentage of receptors showing restricted movement in non stimulated cells is due to such oligomerizations? CCR2 or other receptors would not be fluorescence labeled and could therefore not be detected.

Any interaction with a partner is likely to influence the movement of the receptor. It is therefore not excluded that receptor heterodmerization can slow down CCR5 motion. In our study, we used HEK293 cells, which do not express CCR2 at their surface (no functional response upon activation, doi: 10.1189/jlb.0802415; doi: 10.1126/scisignal.aai8529). The influence of CCR5/CCR2 heterodimerization is not relevant here.

2) The clustering of arrestin has been documented and discussed in the two following manuscripts:1) Coordinate-based co-localization-mediated analysis of arrestin clustering upon stimulation of the C-C chemokine receptor 5 with RANTES/CCL5 analogues.Tarancón Díez L, Bönsch C, Malkusch S, Truan Z, Munteanu M, Heilemann M, Hartley O, Endesfelder U, Fürstenberg A. Histochem Cell Biol. 2014 Jul;142(1):69-77. doi: 10.1007/s00418-014-1206-1. Epub 2014 Mar 13. PMID: 246230382) Quantitative morphological analysis of arrestin2 clustering upon G protein-coupled receptor stimulation by super-resolution microscopy.Truan Z, Tarancón Díez L, Bönsch C, Malkusch S, Endesfelder U, Munteanu M, Hartley O, Heilemann M, Fürstenberg A. J Struct Biol. 2013 Nov;184(2):329-34. doi: 10.1016/j.jsb.2013.09.019. Epub 2013 Sep 30. PMID: 24091038These should be cited and the results being included in the discussion.

We have added the two references and briefly discussed them page 28.

Indeed, a novelty of our article is that even though arrestins can contribute to its own clustering, this is not obverved when the receptor cannot dimerize. The clustering of arrestins would therefore be directly dependent on the structural properties of the receptor. This clustering and the resulting immobilization lead to the optimization of receptor endocytosis.